# Spatial and seasonal effects on the delayed ionospheric response to solar EUV changes

Erik Schmölter[1], Jens Berdermann[1], Norbert Jakowski[1], and Christoph Jacobi[2]

[1]German Aerospace Center, Kalkhorstweg 53, 17235 Neustrelitz, Germany
[2]Leipzig Institute for Meteorology, Universität Leipzig, Stephanstr. 3, 04103 Leipzig, Germany

**Correspondence:** Erik Schmölter (Erik.Schmoelter@dlr.de)

**Abstract.** This study correlates different ionospheric parameters with the integrated solar EUV radiation to analyze the delayed ionospheric response, testing and improving upon previous studies on the ionospheric delay. Several time series of correlation coefficients and delays are presented to characterize the trend of the ionospheric delay from January 2011 to December 2013. The impact of the diurnal variations of ionospheric parameters in the analysis at an hourly resolution for fixed locations are discussed and specified with calculations in different time scales and with comparison to solar and geomagnetic activity. An average delay for TEC of $\approx 18.7$ hours and for foF2 of $\approx 18.6$ hours is calculated at four European stations. The difference between northern and southern hemisphere is analyzed by comparisons with the Australian region. A seasonal variation of the delay between northern and southern hemisphere is calculated for TEC with $\approx 5 \pm 0.7$ hours and foF2 with $\approx 8 \pm 0.8$ hours. The latitudinal and longitudinal variability of the delay is analyzed for the European region, and found to be characterized by a decrease in the delay from $\approx 21.5$ hours at 30°N to $\approx 19.0$ hours at 70°N for summer months. For winter months, a roughly constant delay of $\approx 19.5$ hours is calculated. The results based on solar and ionospheric data at hourly resolution and the analysis of the delayed ionospheric response to solar EUV show the seasonal and latitudinal variations. Results also indicate a relation of the ionospheric delay to geomagnetic activity and a possible correlation with the 11-year solar cycle in the analyzed time period.

## 1 Introduction

The solar extreme ultraviolet radiation (EUV) is the dominant source of ionization in the ionosphere. Therefore, the high variability of EUV within the 27-day solar rotation cycle (Lean et al., 2011), the 11-year solar cycle (Fröhlich and Lean, 2004), or within short-term events like solar flares (Berdermann et al., 2018) has a strong impact on the ionosphere. The resulting photoionization, together with photodissociation, recombination, and transport processes, causes different ionospheric variations that may depend on time or location (Rishbeth and Mendillo, 2001). The structure of the ionosphere is dominated by the interaction of different wavelength ranges in the solar spectrum with the respective particle population and composition

| Publication | Delay [d] | Solar flux parameter | Ionospheric parameter |
|---|---|---|---|
| Titheridge (1973) | 1 | F10.7 | TEC |
| Jakowski et al. (1991) | 1-2 | F10.7 | TEC |
| Jakowski et al. (2002) | 1-3 | F10.7 | TEC |
| Afraimovich et al. (2008) | 1.5-2.5 | F10.7, EUV | Global mean TEC |
| Oinats et al. (2008) | 2-4 | F10.7 | NmF2, TEC |
| Zhang and Holt (2008) | 2-3 | F10.7 | Electron density |
| Min et al. (2009) | 2 | F10.7 | Electron density, TEC |
| Lee et al. (2012) | 1-2 | F10.7 | Electron density |
| Jacobi et al. (2016) | 1-2 | F10.7, EUV | Global mean TEC |
| Ren et al. (2018) | 1 | EUV | Electron density |

**Table 1.** The table presents results from former studies, which provide an approximate ionospheric delay to solar activity at a daily resolution.

at specific altitudes. This results in different ionospheric layers defined by the density distribution of the ion species (Kelley, 2009). A detailed understanding of the ionospheric chemical and physical processes is needed to provide realistic and reliable

physics-based models. The delayed ionospheric response to solar EUV radiation is captured in various ionospheric models (Ren et al., 2018; Vaishnav et al., 2018) and respective simulations can confirm results of previous studies estimating the ionospheric delay with observational data at a daily resolution. The calculation of the delay with observational data at higher temporal resolution ($\leq 1$ hour) is of interest, as it permits more detailed descriptions of temporal and spatial variations. The dependence on solar and geomagnetic activity (Ren et al., 2018) can also be explored further. In the future, results for the

ionospheric delay at high temporal resolution will strengthen the understanding of ionospheric processes and help to validate physics-based models.

Former analyses of the ionospheric electron content changes in connection with solar flux variations, in particular on the 27-day rotation time scale, have revealed that ionospheric parameters have a delayed response to solar variability. A selection of these studies is presented in Table 1. In these studies, the ionospheric delay was calculated using different EUV proxies

or measurements of the EUV flux at daily resolutions. The recent results by Ren et al. (2018) from observational and model calculations specified different features of the ionospheric delay. A strong impact of the geomagnetic activity on the ionospheric delay to solar EUV changes was found. Simulations with the Thermosphere Ionosphere Electrodynamics General Circulation Model (TIEGCM) and theoretical calculations were used to discuss the influence of ion production and loss on the ionospheric delay. The impact of the O/$N_2$ ratio on the delay was analyzed as well. The ion production responds immediately to EUV

variations and depends on both the solar EUV flux contribution and the O/$N_2$ ratio. The loss is delayed and controlled by the O/$N_2$ ratio, which in turn is also dominantly controlled by the solar EUV flux contribution. The resulting ionospheric response could be further modulated by dynamic and electrodynamic processes in the ionosphere. In addition, a latitudinal dependence of the ionospheric delay was shown (Ren et al., 2018).

This study analyzes the delay at high temporal resolution of one hour. Furthermore, the hemispheric dependence of the ionospheric delay is examined with a detailed study of the European region. This analysis uses on GNSS and ionosonde data over Europe and Australia. The time series of the delays and the correlation coefficients are calculated between solar EUV radiation and two ionospheric parameters: the Total Electron Content (TEC) and the critical frequency of the F2 layer (foF2). TEC measured the vertical integrated electron density and can be used to describe changes in the whole ionosphere-plasmasphere system due to solar EUV variability. The variations of TEC are mostly controlled by the F2 layer (Lunt et al., 1999; Petrie et al., 2010; Klimenko et al., 2015) and for mid-latitudes the total plasmaspheric contribution to TEC is between approximately 8 to 15 % during daytime and approximately 30 % during nighttime (Yizengaw et al., 2008). The availability of TEC in maps with good data coverage for certain regions (e.g. European or North American region) allows a spatial analysis of the delay and a comparison with the foF2 data for specific locations. On the other hand, foF2 describes only the F2 layer of the ionosphere without complicating contributions from the plasmasphere and lower ionospheric layers. Both ionospheric parameters are highly correlated (Kouris et al., 2004), but variations like different peak time of the diurnal variation (Liu et al., 2014) could have a considerable impact on the delayed ionospheric response. As expected, the results will show that the ionospheric delay is very similar for TEC and foF2.

## 2 Data

### 2.1 Solar EUV raidation

Parts of the EUV spectrum has been continuously measured since 2000 C.E., with publicly available EUV observations provided by the Solar EUV Experiment (SEE) onboard the Thermosphere Ionosphere Mesosphere Energetics and Dynamics (TIMED) satellite (Woods et al., 2005), the Geostationary Operational Environmental Satellites (GOES) (Machol et al., 2016) and the Solar Auto-Calibrating EUV/UV Spectrophotometers (SolACES) (Nikutowski et al., 2011; Schmidtke et al., 2014). The data used in this paper are from the Solar Dynamics Observatory (SDO) EUV Variability Experiment (EVE) (LASP, 2019). They represent almost the entire EUV spectrum, with a wavelength range from 0.1 to 105 nm, a spectral resolution of 0.1 nm and a temporal resolution of 20 s. The EUV data cover several years (2011 to 2014) without large data gaps (Woods et al., 2012).

### 2.2 Ionospheric parameters

The analysis correlates EUV with two important ionospheric parameters, appropriate to investigate features of the ionospheric delay. The first parameter is TEC, which is an integral measurement of the electron density and well suited for the analysis of the ionospheric response to solar EUV variations. The parameter was used in several preceding studies to calculate the ionospheric delay (see Table 1). The time series of TEC for single locations and regions is extracted from the International GNSS Service (IGS) TEC maps (NASA, 2019b), which provide global coverage since 1998 at the required high resolution of at least one hour (Hernández-Pajares et al., 2009). These TEC data represent a weighted average between real observations and

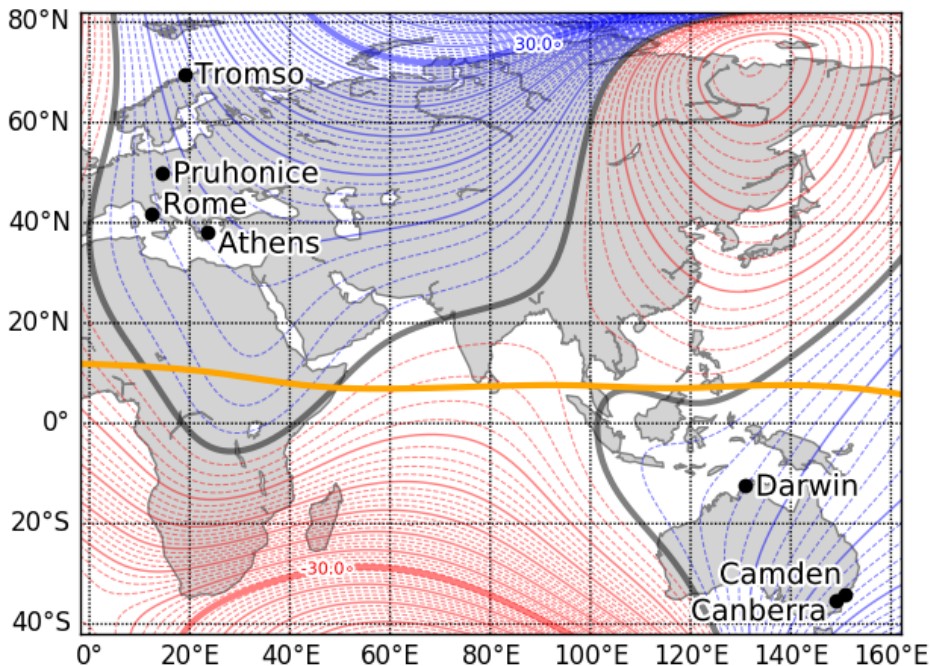

**Figure 1.** The European (Tromsø, Průhonice, Rome and Athens) and Australian (Darwin, Camden, Canberra) ionosonde stations which are used in the calculation of the delayed response of the ionosphere to solar EUV variations. Earth's magnetic field is presented with the geomagnetic equator (orange line) and the magnetic declination (blue, red and black lines) from the World Magnetic Model (NASA, 2019c).

an ionospheric model, dependent on the availability of observations at a given time and location. The chosen IGS TEC maps by Universitat Politècnica de Catalunya (UPC) use a global voxel-defined 2-layer tomographic model solved with Kalman filter and spline interpolation (Orús et al., 2005; Hernández-Pajares et al., 2016). In preparation for the delay calculation, TEC values at seven ionosonde locations and one region (Europe) were extracted from the IGS TEC maps. For each ionosonde location the nearest grid point in the maps was used.

The other ionospheric parameter included in the analysis, foF2, is derived from ionosonde station data (NOAA, 2019) provided by the National Oceanic and Atmospheric Administration (NOAA), and are available for the same time periods at temporal resolution of 15 minutes (Wright and Paul, 1981). Figure 1 shows a map of stations used to calculate the ionospheric delay. The geographic and geomagnetic latitudes and longitudes of the stations are shown in Table 2. In the northern hemisphere, the European stations Tromsø, Průhonice, Rome, and Athens were derived from auto-scaled ionosonde, since they cover different

latitudes ranging from $\approx 38°N$ to $\approx 70°N$. The dense coverage of GPS stations over Europe allows a good comparison with TEC data for these locations (Belehaki et al., 2015). An analysis of the southern hemisphere with the South African region would be preferred because of a similar longitude, but there are some time and data gaps, which prevented a reliable estimation of the delay for the available stations. Instead, auto-scaled data from the Australian stations Darwin, Camden, and Canberra are

| Station | geographic [°] | | geomagnetic [°] | | magnetic [°] | |
|---|---|---|---|---|---|---|
| | Lat. | Lon. | Lat. | Lon. | Dec. | Inc. |
| Tromsø | 69.7 | 19.0 | 67.2 | 115.9 | 7.0 | 78.2 |
| Průhonice | 50.0 | 14.6 | 49.3 | 98.6 | 2.9 | 65.9 |
| Rome | 41.8 | 12.5 | 41.8 | 93.6 | 2.2 | 58.0 |
| Athens | 38.0 | 23.6 | 36.2 | 103.3 | 3.7 | 54.5 |
| Darwin | -12.4 | 130.9 | -21.5 | -155.7 | 3.3 | -39.7 |
| Camden | -34.0 | 150.7 | -40.1 | -131.6 | 12.4 | -64.5 |
| Canberra | -35.3 | 149.0 | -42.3 | -133.2 | 12.3 | -66.0 |

**Table 2.** Geographic and geomagnetic latitudes and longitudes of the European (Tromsø, Průhonice, Rome and Athens) and Australian (Darwin, Camden, Canberra) ionosonde stations which are used in the calculation of the delayed response of the ionosphere to solar EUV variations. The magnetic declination and inclination are shown as well. The magnetic field parameters are calculated with the International Geomagnetic Reference Field (NASA, 2019d).

used for the analysis in the southern hemisphere. These stations cover latitudes between $\approx 12°$S to $\approx 35°$S. The conditions of Earth's magnetic field for the European and Australian stations are comparable, with a small magnetic declination and similar absolute value of magnetic inclination (see Table 2). The selected stations seem appropriate for a comparison between northern and southern hemisphere due to these similar conditions. The variability of the characteristic ionosphere parameter, foF2, measured with ionosondes are compared to the EUV flux. In preparation of the analysis, all data are resampled to an hourly resolution using the mean foF2. Gaps are filled with a linear interpolation. Delay calculations during data gaps of several days do not succeed due to the lack of a defined peak in the cross-correlation. This causes corresponding gaps in the observed trend of the ionospheric delay. Unlike in Schmölter et al. (2018), there are no band-stop filters used to reduce the daily variations, since this calculation step does not add more reliability to the delay calculations. The Kp-index (NASA, 2019a) is used to characterize the influence of the geomagnetic activity on the delay in the analysis.

## 3   Correlation of ionospheric parameters with solar EUV

The delayed ionospheric response to solar variability was calculated by different studies at daily resolution. A selection of these studies are shown in Table 1. The first delay calculation with cross-correlations at an hourly resolution was performed by Schmölter et al. (2018). This work extends the previous research by addressing daily, seasonal, and regional dependencies of the ionospheric delay at a high temporal resolution. The analysis compares the ionospheric delay in the TEC and foF2 from different locations. Their corresponding time series are examined for different temporal variations, including: diurnal, 27-day solar rotation cycle, and seasonal. Figure 2 shows the impact of the diurnal variations on the correlation coefficients by comparing different temporal resolutions (weekly, daily, and hourly). The hourly resolution TEC data are extracted from IGS TEC maps (NASA, 2019b) at Rome ($41.8°$N and $12.5°$E). The EUV data are integrated SDO-EVE fluxes from 6 to

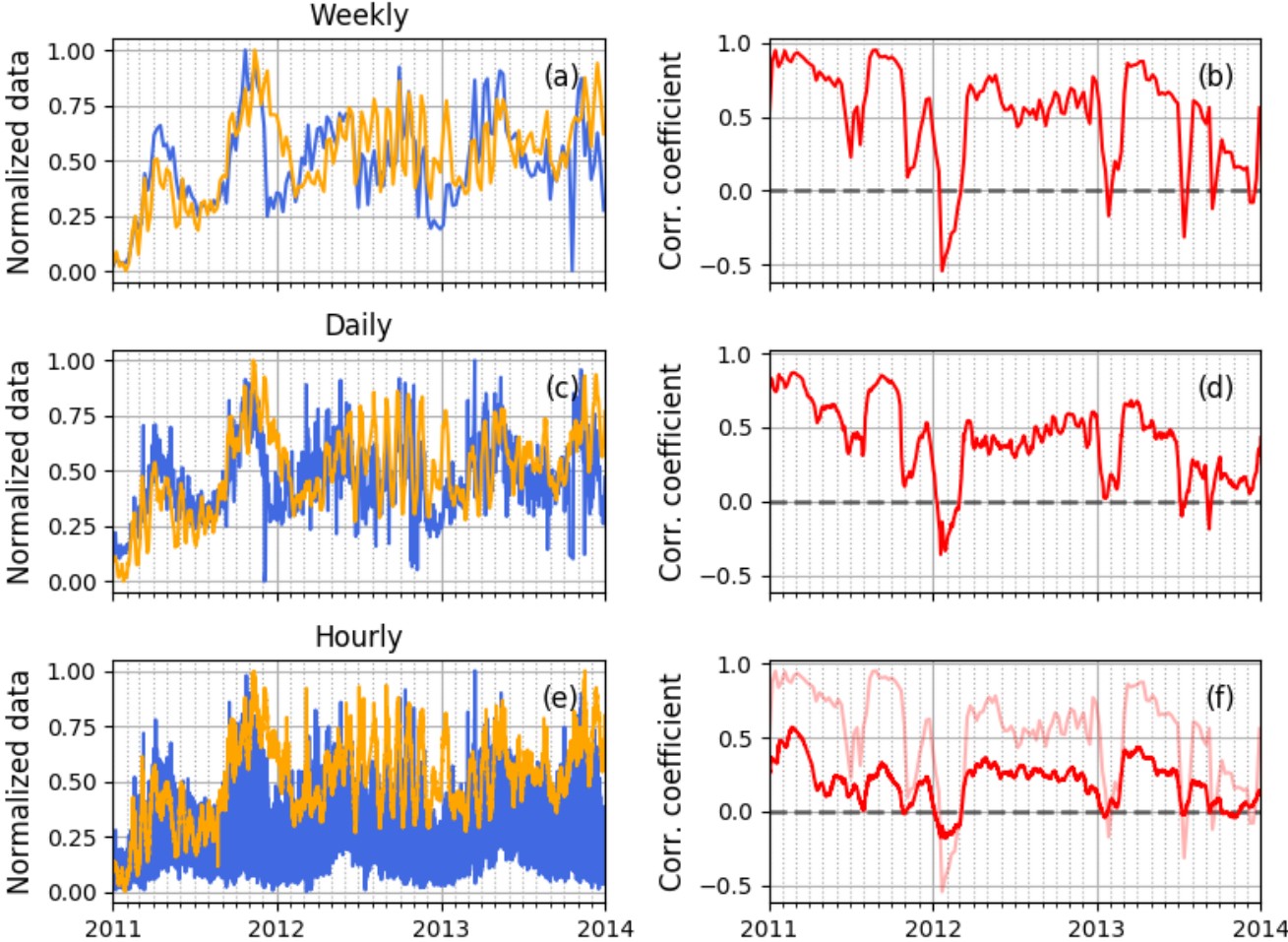

**Figure 2.** The plots show the normalized TEC (blue) and EUV (orange) data, as well as the resulting correlation coefficients (red), for different temporal resolutions: weekly (a, b), daily (c, d) and hourly resolution (e, f). The correlation coefficients were calculated using a time window of approximately 90 days and a step size corresponding to each resolution. The daily and weekly TEC data were retrieved by calculating the mean for the values from 11:00 to 13:00 local time each day. The correlations coefficients for the weekly resolution are shown in the plot for the hourly resolution again (light red). All data correspond to the location of Rome with $41.8°$N and $12.5°$E.

nm (LASP, 2019). The daily and weekly data sets for TEC are retrieved by calculating the corresponding means for the values from 11:00 to 13:00 local time each day, i.e. only the time periods with an expected maximum photoionization are 110 considered. The correlation coefficients between EUV and TEC data are calculated using a time window of approximately 90 days. The comparison of correlation coefficients in hourly and weekly resolution in Figure 2 shows that the correlation at hourly resolution is, as expected, much smaller. Increases and decreases of the correlation coefficients have the same trend,

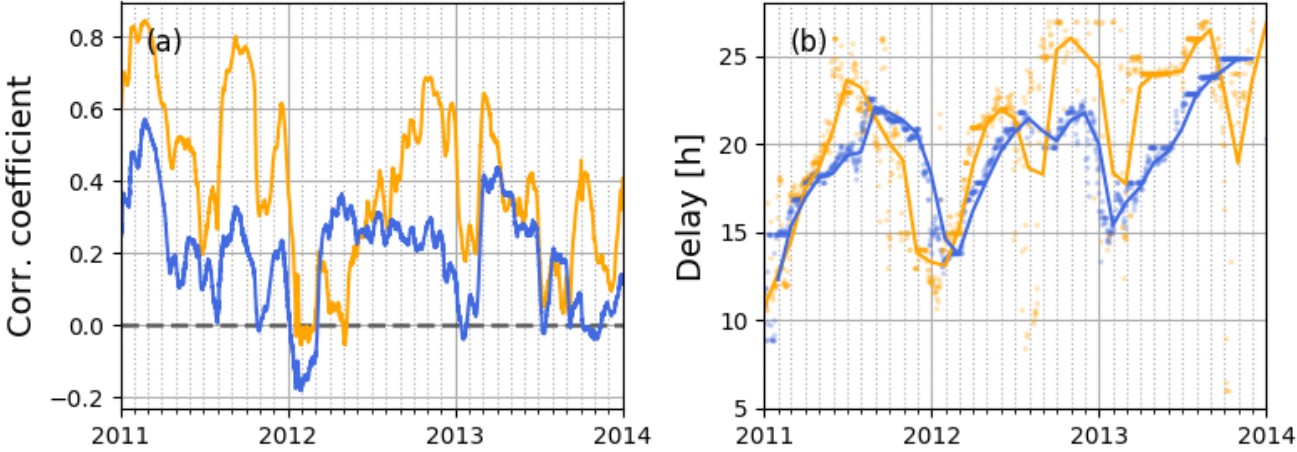

**Figure 3.** Plot (a) shows the correlation coefficients and plot (b) the delays calculated with a fixed location (blue) and a fixed local time (orange). The fixed location is Rome (41.8°N and 12.5°E) and the fixed local time is 12:00 at 40°N. The correlation coefficients and delays were calculated using a time window of approximately 90 days and a step size of 1 hour with TEC and EUV data. The delays at hourly resolution are shown by dots and the monthly means of the delays are shown as solid lines.

though. A characterization of the correlation trend is possible in all shown resolutions. The varying correlation between solar EUV flux or solar proxies like F10.7 with TEC is expected from preceding studies.

Solar EUV radiation does not fully control the ionospheric variability at all time periods and on all time scales, resulting in the low correlation coefficients shown in 2 (b), (d), and (f) (Unglaub et al., 2012). The magnitude of the correlation coefficient has been shown to relate to the strength of the impact of other processes (Verkhoglyadova et al., 2013). Analyzing times of both high and low correlation between solar EUV flux and ionospheric parameters is important to understand the changes in ionospheric processes and interactions.

In Figure 3 the correlation coefficients and delay between TEC and EUV are shown for a fixed location (Rome with a latitude of 41.8°N and a longitude of 12.5°E) and a fixed local time (12:00) at the same latitude (40°N). The correlation coefficients and delay for both results are calculated with cross-correlations using a time window of approximately 90 days for the TEC and EUV data. The two methods differ only in the way that the TEC time series was extracted from the TEC maps. For the calculation with fixed location, the latitude and longitude are unchanged for each data point. For the calculation with fixed

local time, the longitude is changed to correspond with the location at 12:00 local time. In Figure 3 the differences in the correlation coefficients are shown. The correlation coefficients for a fixed local time are greater than for a fixed location, but strong increases or decreases of the trend appear in both data sets (e.g. the strong decreases in the end of 2011 and 2012). The trend of the delay with a slight increase over the three years as well as the annual variation are present. The two different approaches have a mean variance of approximately 3.15 hours, which accounts for an uncertainty of approximately 16.04

% in the ionospheric delay calculation. This is an acceptable impact of the diurnal variation on the trend of the delay for characterizing temporal and spatial changes.

The delayed ionospheric response to solar EUV radiation depends on the solar local time, and the calculated results for fixed locations can be understood as a mean ionospheric delay for different local times. This makes the fixed local time approach preferable for further analysis. However, its utility is limited since the time series extracted from the IGS TEC maps rely less

on measurements (and more heavily on the background model) when considering areas with few or no ground stations. Thus, this study preferentially utilizes the fixed location method, since a location with good data coverage is more easily selected. And despite the strong diurnal variations in the ionospheric parameters and their impact on both the correlation and the delay calculations, Figures 2 and 3 show that relative trends can be calculated at hourly resolutions for fixed locations. The significant decreases of the correlation and the negative correlation coefficients are not effects of the diurnal variations, since they are of

the same order for all results and the observed trend must have another origin (see Figures 2 and 3).

Geomagnetic activity and thermospheric conditions also impact the ionospheric state. The period of this study (January 2011 through December 2013) covers the ascending phase and beginning of the main phase of the 24th solar cycle. The geomagnetic activity during this time is on very low levels compared to previous ascending phases with geomagnetic storm rates that compare to solar minima in previous cycles (Richardson, 2013). The solar activity of the cycle is also significantly

lower compared to previous cycles and a much weaker ionization of the ionosphere occurs (Hao et al., 2014). These complex variations are not covered by EUV flux measurements and cannot be characterized with the cross-correlations between solar EUV and ionospheric parameters. In Figure 4 the calculated correlation coefficients and delays from the location Rome (already shown in Figure 3) are compared to the Kp-index as a measure of the geomagnetic activity. The smoothed trends of the Kp-index, correlation coefficient between EUV and TEC as well as delay between EUV and TEC show similar decreases in all

three data sets during the end of each year. The minimum of the correlation coefficient and the delay are about two month behind the minimum of the Kp-index. For the northern hemisphere the comparison of the Kp-index with the gradient of the delay in Figure 5 shows clear correlations for each year ($\approx 0.53$ in 2011, $\approx 0.70$ in 2012 and $\approx 0.77$ in 2013) indicating that the geomagnetic activity modulates the ionospheric delay. For the southern hemisphere the comparison of the Kp-index with the gradient of the delay in Figure 6 shows good correlations in the first two years ($\approx 0.80$ in 2011, $\approx 0.73$ in 2012 and $\approx -0.01$

in 2013). There is no correlation in 2013, which is due to strong deviations of the calculated ionospheric delay in the end of the year. The strong impact of the geomagnetic activity on the delay was reported e.g. by Ren et al. (2018), and Figures 4, 5 and 6 give a first indication about such a relation. The trend appears in both hemisphere for mid-latitudes indicating a global trend. The analysis will show more results for both hemisphere to confirm the observed relation.

In conclusion, the results in Figures 2, 3 and 4 show that the diurnal variations have an impact on the correlation between

EUV and TEC at hourly resolution. There is no significant changes in the trend and the information about different variations can be retrieved. The following analysis will characterize certain variations at longer time scales, while keeping in mind that their magnitude may differ due to the deviations caused by the diurnal variations.

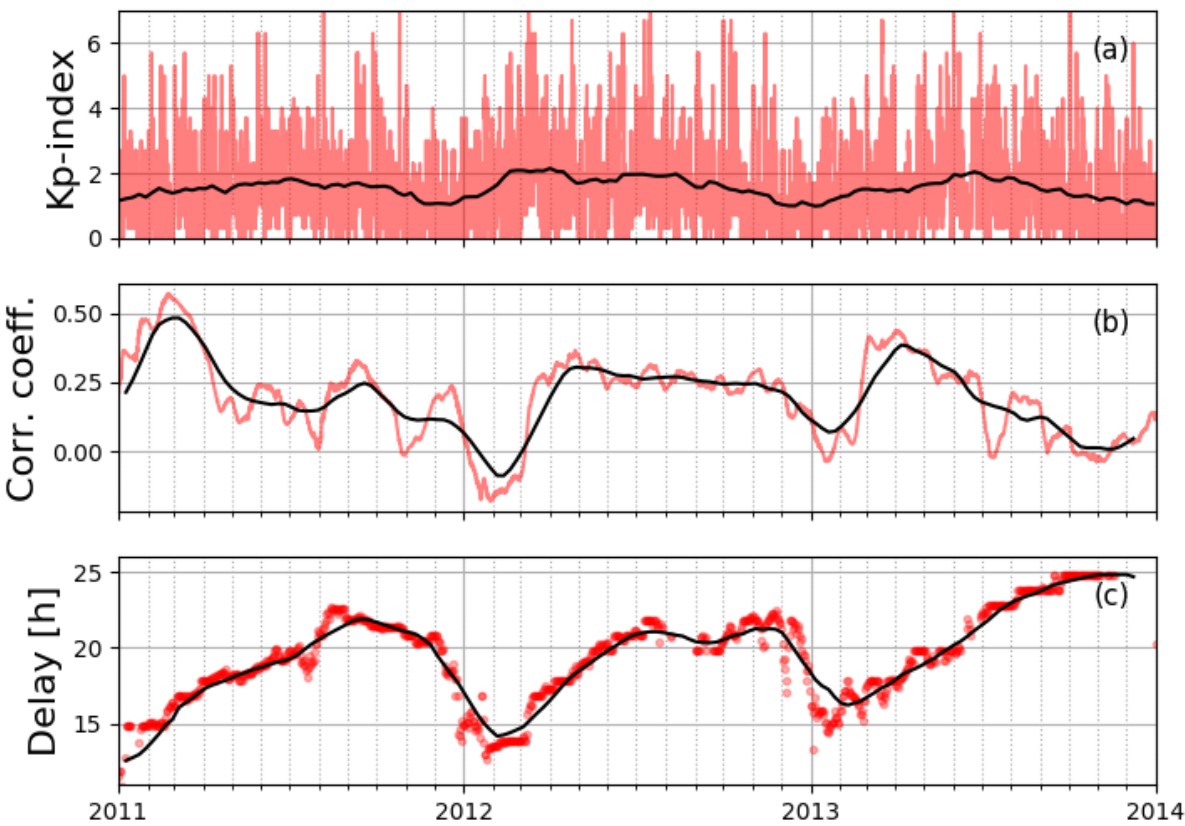

**Figure 4.** The transparent red lines or dots show the raw data: Kp-index (a), correlation coefficients between EUV and TEC (b) and delays between EUV and TEC (c) at hourly resolution. The black lines show the smoothed weekly means to present the overall trend (running mean with window size of 10 days). All data correspond to the location of Rome with 41.8°N and 12.5°E.

## 4 Representation of the delay for TEC and foF2

In earlier studies, the correlation of the ionospheric delay has been calculated for different ionospheric parameters based on daily at hourly resolutions, as shown in Table 1. For example, Jakowski et al. (1991) used the solar radio flux index F10.7 and calculated a delay of one to two days. Jacobi et al. (2016) confirmed this delay with satellite-based EUV-TEC measurements (Unglaub et al., 2011) and also calculated the ionospheric delay with EUV fluxes. The validation with EVE flux measurements was important because the solar rotation variations of F10.7 and EUV are not synchronized at all times and the calculated ionospheric delay with F10.7 might be greater than the actual delayed ionospheric response to EUV (Woods et al., 2005; Chen

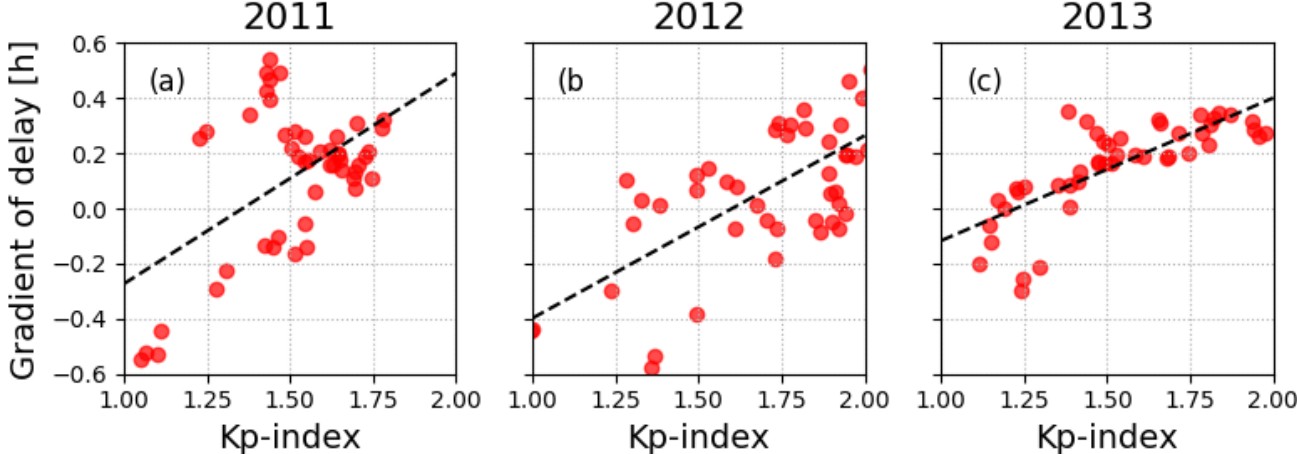

**Figure 5.** The scatter plots for 2011 (a), 2012 (b) and 2013 (c) show the correlation between the Kp-index and gradient of the delay. The smoothed weekly means (running mean with window size of 10 days) are used for this comparison. Correlation coefficients of $\approx 0.53$ (a), $\approx 0.70$ (b) and $\approx 0.77$ (c) are estimated. All data correspond to the location of Rome at $41.8°$N and $12.5°$E.

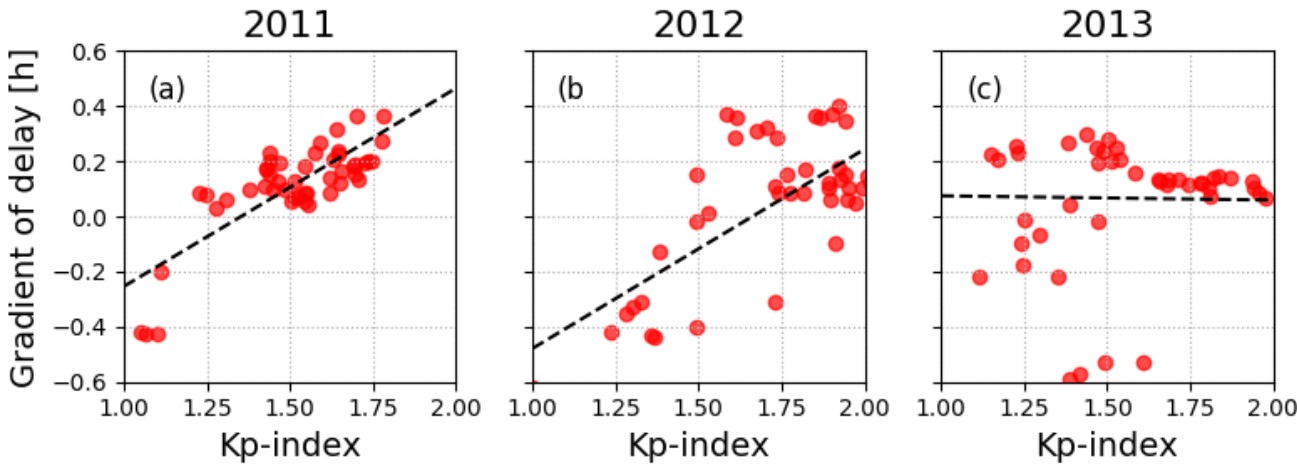

**Figure 6.** The scatter plots for 2011 (a), 2012 (b) and 2013 (c) show the correlation between the Kp-index and gradient of the delay. The smoothed weekly means (running mean with window size of 10 days) are used for this comparison. Correlation coefficients of $\approx 0.53$ (a), $\approx 0.70$ (b) and $\approx 0.77$ (c) are estimated. All data correspond to the location of Canberra at $35.3°$S and $149.0°$E.

et al., 2018). Schmölter et al. (2018) used EVE and GOES EUV fluxes to calculate an ionospheric delay of about 17 hours as mean value based on data at hourly time resolution.

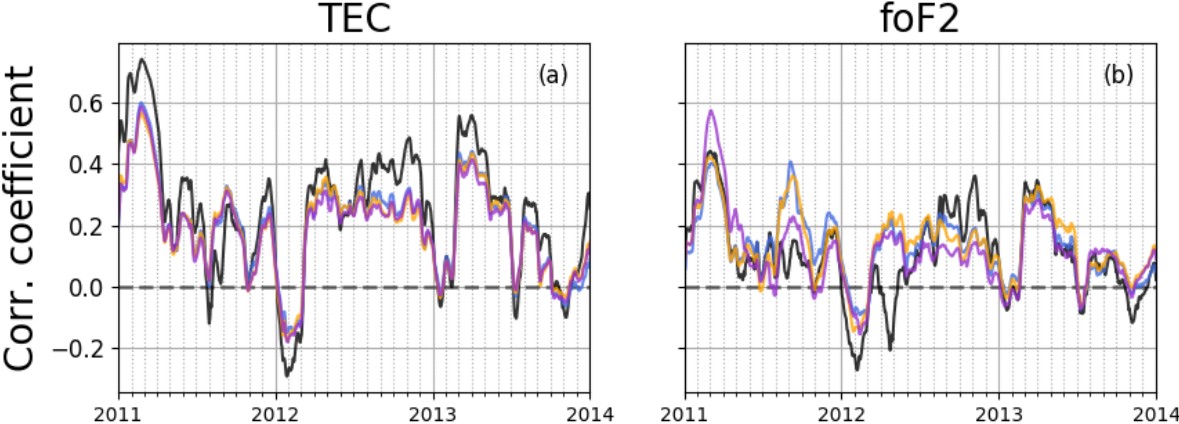

**Figure 7.** The plots show the correlation coefficients of the ionospheric parameters TEC (a) and foF2 (b) with integrated EVE fluxes (6 to 105 nm) for Tromsø (black), Průhonice (blue), Rome (orange), and Athens (purple). All parameters were analyzed at hourly resolution using a time window of 90 days and a step size of one hour.

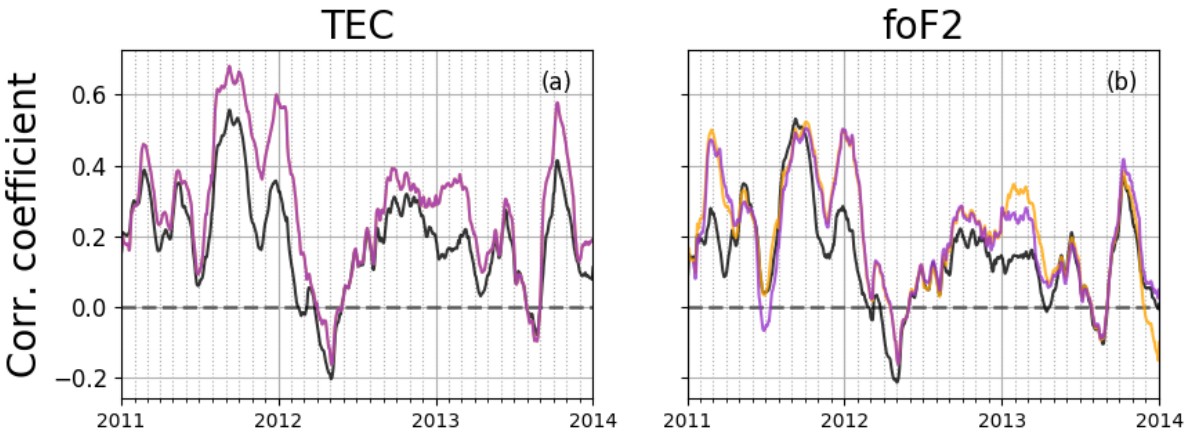

**Figure 8.** The plots show the correlation coefficients of the ionospheric parameters TEC (a) and foF2 (b) with integrated EVE fluxes (6 to 105 nm) for Darwin (black), Camden (orange), and Canberra (purple). All parameters were analyzed at hourly resolution using a time window of 90 days and a step size of one hour.

In the calculation of the ionospheric delay, a time window of 90 days and a step length of one hour are used for the cross-correlations. This time frame not only allows to produce reliable results for the delay, it also allows to identify changes in the ionospheric processes at this location. The calculation is applied to the time series from December 2010 to February 2014 and covers a time period of roughly three years.


The results for the European stations are shown in Figure 7 for TEC and foF2. The trend of the correlation coefficients of TEC for the four European stations are very similar. The station Tromsø has more significant peaks (for increases and decreases in the correlation), but follows the same general trend. At the end of each year the correlation decreases significantly and reaches negative values. In Figure 4, this was interpreted as a possible effect of geomagnetic activity. At the end of the chosen time period, the correlation coefficient drops due to data gaps and the applied interpolation method.

The correlation coefficients of foF2 for the four European stations are smaller than those of the TEC. However, the trends of the two correlation coefficients are similar for the different stations. The correlation coefficients for the station Tromsø again show the largest deviation from the mean of the trends of all stations. Since Tromsø is an auroral station, the processes in the ionosphere for this location are influenced by other mechanisms, e.g., particle precipitation or thermospheric heating controlled by the solar wind (Hunsucker and Hargreaves, 2002). In this study, the station at Tromsø provides a high-latitude boundary for the analysis of the delayed ionospheric response in the European region.

The TEC and foF2 correlation coefficients for the Australian stations are shown in Figure 8. In general, the TEC and foF2 correlation coefficients at the Australian stations are slightly larger than the corresponding correlation coefficients at the European stations. The trend of correlation coefficients for both parameters and the trend for the different stations are in good agreement. The suggested impact of the geomagnetic activity is less present in these results. Most notably, the decrease and minimum in December 2012 does not occur. The difference might be due to further impacts on the correlation, e.g. thermospheric wind conditions or seasonal variations due to composition changes (atomic/molecular ratio), which are not covered in this study, but are known to have a strong impact on the ionospheric state (Rishbeth, 1998; Rishbeth et al., 2000).

The results of the delay calculation through cross-correlations are shown in Figure 9 and 10. The trend of the delay for TEC and foF2 at the European stations in Figure 9 is in agreement with the trend found by Schmölter et al. (2018), having a slow increase in the delay during the first half of the year, a maximum of the delay close to the end of the year and a sudden decrease of the delay at the end of the year. This pattern repeats in the three years of the chosen time period. The trend of the delay for TEC and foF2 at the Australian stations in Figure 10 is very similar, but shows a less linear increase of the delay in each year. Contrary to the correlation coefficients in the Figures 7 and 8, the delays show a good correlation with the geomagnetic activity in both hemispheres. Hence, this global trend confirms an additional dependence of the delay on the geomagnetic activity.

The maxima of the delay increase from year to year in 2011 to 2013 (especially for foF2) in the northern hemisphere. A similar trend occurs in the southern hemisphere from 2011 to 2012. This small increase might result from the growing solar activity in the same time period. Figure 11 shows the data for integrated EUV during the analyzed time period and the calculated delay for TEC at Rome and Canberra. As a very coarse visualization for the correlation between EUV and delay, the linear trends in both data sets are shown as well. The long-term trends of EUV and the delay on the northern and southern hemisphere increase within the chosen time period. Thus, during the solar maximum (cycle 24), long-term changes in the EUV seem to correlate with variations in the delay. A similar behavior was suggested by Schmölter et al. (2018) based on an analysis using GOES data for the same time period. Rich et al. (2003) indicated a smaller delay for solar minimum and a longer delay for solar maximum, and Chen et al. (2015) found a decrease in the trend of the delay with decreasing solar activity. Both analyses calculated the delay at a daily resolution for longer time periods than the one used in this study.

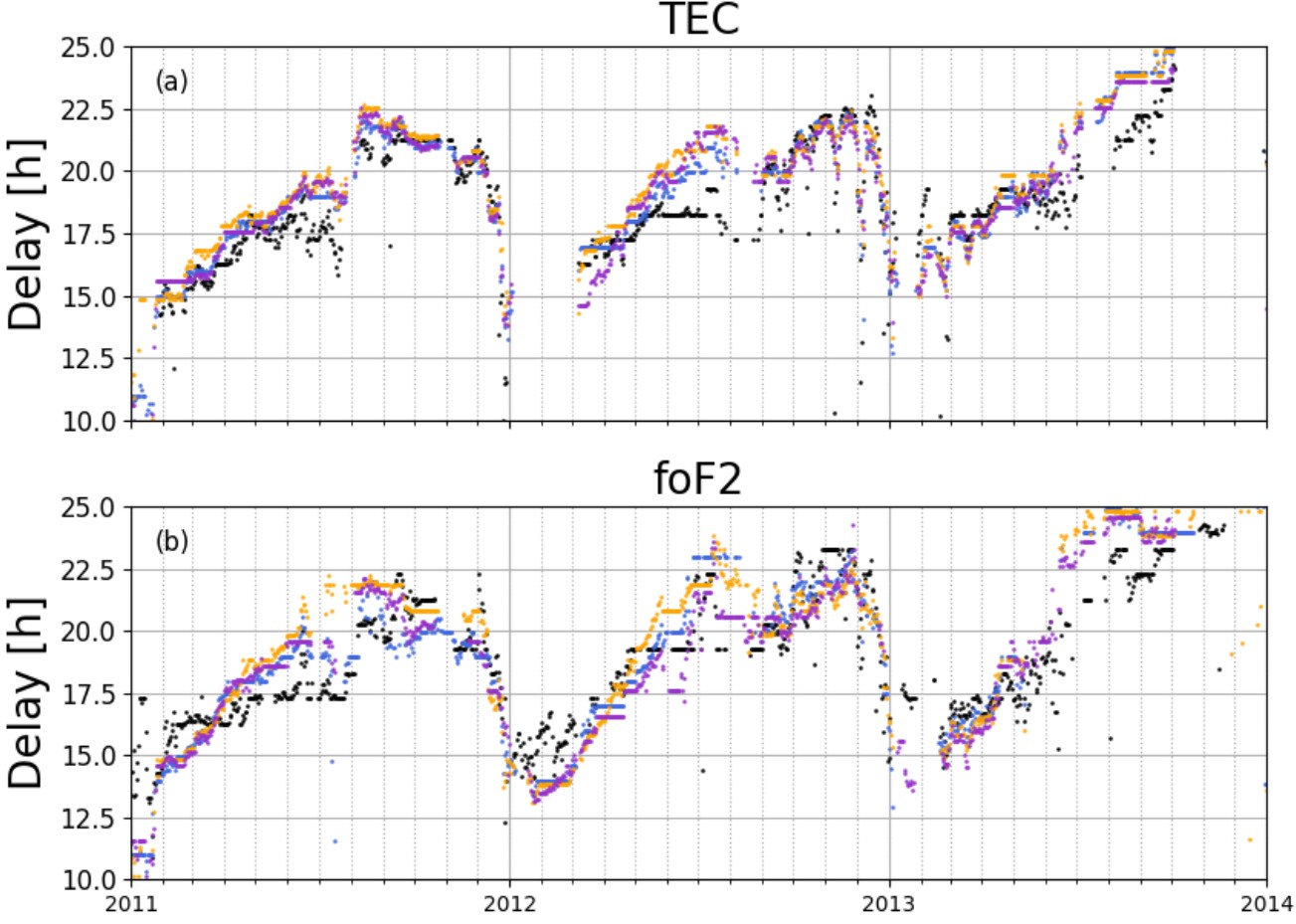

**Figure 9.** The plots show the delays of the ionospheric parameters TEC (a) and foF2 (b) with integrated EVE fluxes (6 to 105 nm) for Tromsø (black), Průhonice (blue), Rome (orange), and Athens (purple). All parameters were analyzed at hourly resolution using a time window of 90 days and a step size of one hour.

The difference between the ionospheric delay for the European and Australian stations in Figures 7 and 8 show only small differences due to the assumed trend with the geomagnetic activity. This trend has to be removed in the further analysis. Therefore, the European station Rome with a latitude of 41.8°N (geomagnetic latitude 41.8°N) and the Australian station Canberra with a latitude of 35.3°S (geomagnetic latitude 42.3°S) are used for the comparison of the northern and southern hemispheres. 215   The non-seasonal trends are removed by calculating the difference between the ionospheric delays of both stations. The results are shown in Figure 12. The difference between both stations clearly shows a seasonal variation in the northern and southern hemisphere with a greater delay for Rome in the northern hemisphere summer and a greater delay for Canberra in the southern hemisphere summer. The delay difference varies over different ranges for the parameters: TEC with $\approx 5 \pm 0.7$ hours

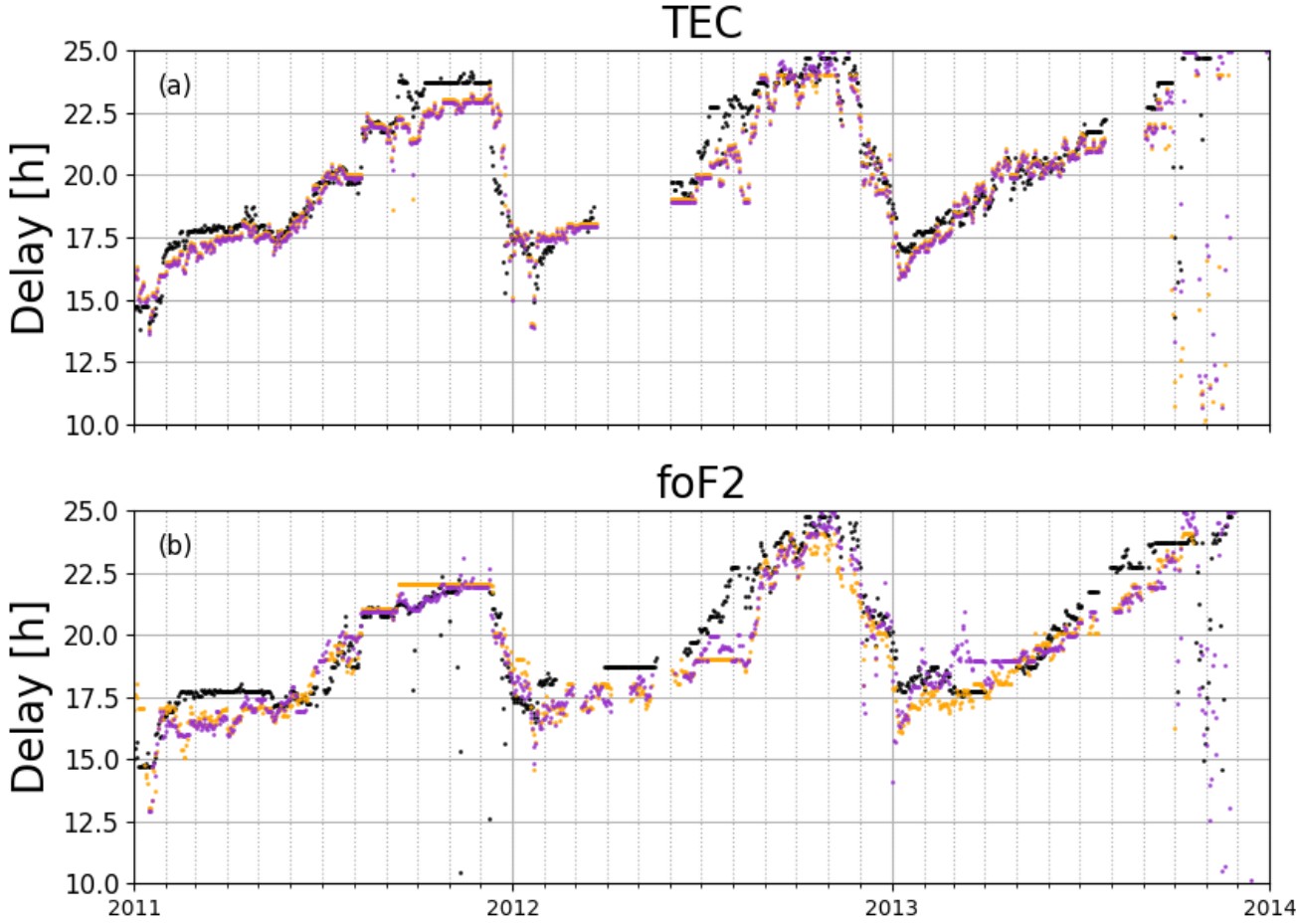

**Figure 10.** The plots show the delays of the ionospheric parameters TEC (a) and foF2 (b) with integrated EVE fluxes (6 to 105 nm) for Darwin (black), Camden (orange), and Canberra (purple). All parameters were analyzed at hourly resolution using a time window of 90 days and a step size of one hour.

and foF2 with $\approx 8 \pm 0.8$ hours. These results could indicate a stronger seasonal variation of the ionospheric delay in the F2
layer compared to the whole ionosphere-plasmasphere system, but there are other possible sources for the difference (e.g. the background model of the IGS TEC maps). Similar to the discussion of the impact of diurnal variations, such findings need to be confirmed with modeling efforts. In conclusion, the trends of the ionospheric delay for TEC and foF2 are very similar and both ionospheric parameters show features of the seasonal variations.

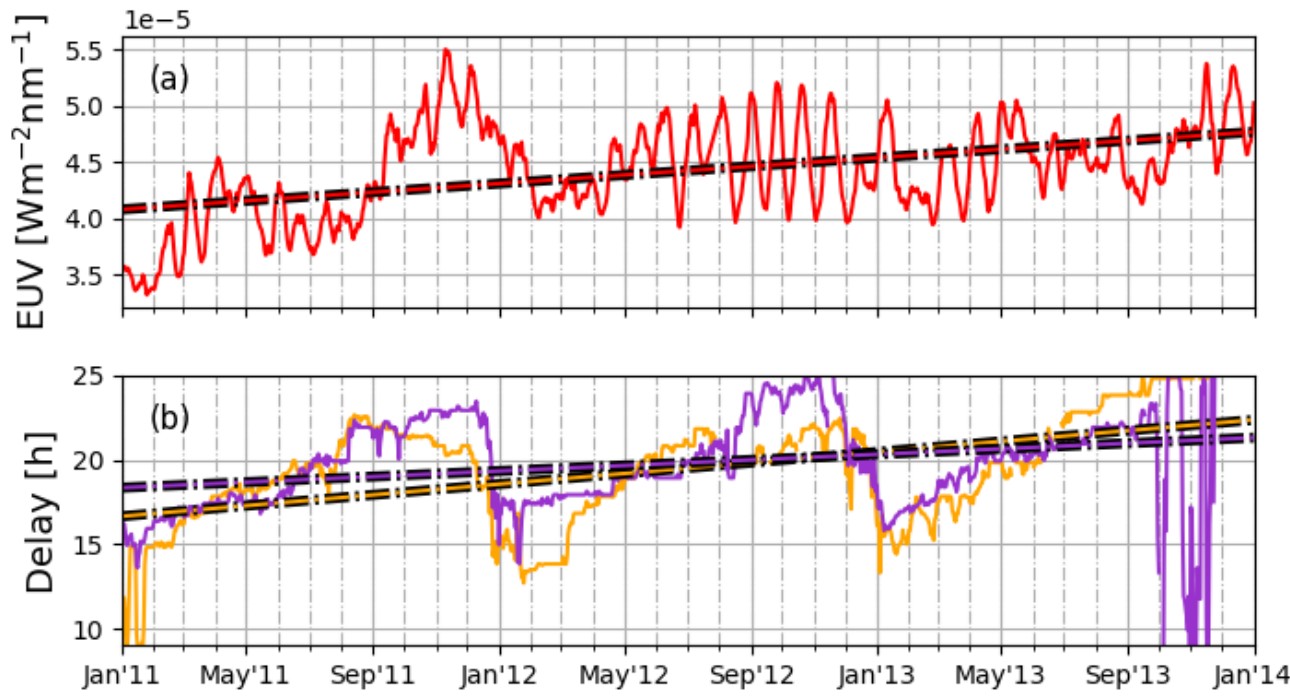

**Figure 11.** Plot (a) shows the the integrated EUV fluxes from 6 to 105 nm and the linear trend of the EUV (dash-dotted line). Plot (b) shows the delays of TEC against EUV for Rome (orange) and Canberra (purple), as well as the linear trends of the delays (dash-dotted lines).

## 5 Analysis of the delay for mid-latitudes

Another trend visible in Figure 9 is a decrease of the delay with latitude in summer. The station at Tromsø shows the shortest delay of the European stations for both parameters. The differences in the delay between Průhonice, Rome, and Athens are smaller. Figure 13 shows the difference between the stations Rome and Tromsø for both ionospheric parameters. The results for TEC show a greater or similar ionospheric delay for the station Rome compared to the station Tromsø. There are only a few short time periods during winter with a greater ionospheric delay for the station Tromsø. A stronger seasonal variation

appears for the parameter foF2, but overall the ionospheric delay is still greater for the station Rome. The mean difference for results in Figure 13 is $\approx 1.08$ hours for TEC and $\approx 0.52$ hours for foF2. The changes with latitudinal dependence of the trends during winter are due to the stronger increase of the ionospheric delay for Rome during summer. No such trend is visible for the Australian stations and there are only minimal differences in the delay. This is probably due to the smaller range of latitudes covered by this stations. A precise interpretation of the trend without data from different latitudes in the southern hemisphere

is difficult. Nonetheless, the results for the latitudes over Europe are consistent with the expectations that different and more varying delays can be observed in polar regions due to the direct impact of the solar wind (Watson et al., 2016) as well as for the equatorial region due to the strong dynamics in ionosphere and thermosphere (Maruyama, 2003).

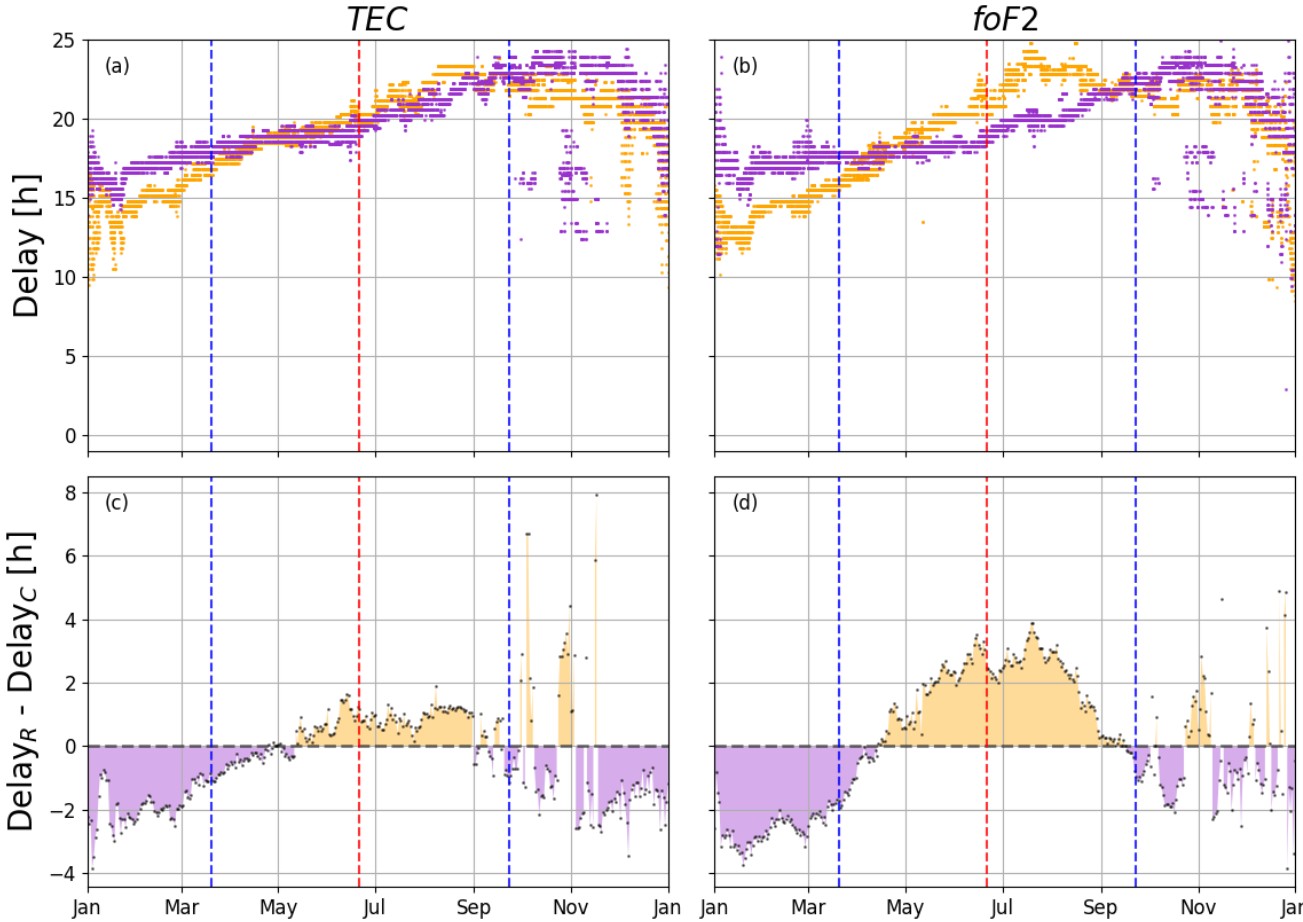

**Figure 12.** Superposed epoch plots for the delay (a, b) and difference in delays (c, d) for the ionospheric parameters TEC and foF2 with integrated EVE fluxes (6 to 105 nm) for Rome (orange) and Canberra (purple). The temporal resolution is one hour. Equinoxes are marked with the blue dashed lines and solstice is marked with the red dashed line.

A further analysis of the mid-latitude delay is possible using TEC data over Europe, where good observational coverage from GNSS stations and minimal influence by the ionospheric model is expected. Therefore, the region from the TEC maps (30°N to 70°N and 10°W to 30°E) can be extracted and the time series of the delay for each available grid point can be calculated. This was done by calculating cross-correlations with a time window of 90 days and a step length of one hour, as shown in Figure 14, which maps the mean delay values for the mid-latitudes in summer (May-August) and winter (November-February). Figure 14 shows ionospheric delays that are consistent with the results from the European ionosonde stations in Figure 9. In winter, there is no strong increase or decrease with latitude, but roughly the same delay of $\approx 19.5$ hours over the entire region. The decrease of the ionospheric delay at latitudes greater than 65°N and smaller 35°N confirms a latitudinal trend, which was found in preceding studies (Lee et al., 2012). A similar behavior of the delay has been found by Ren et al. (2018). In summer, the

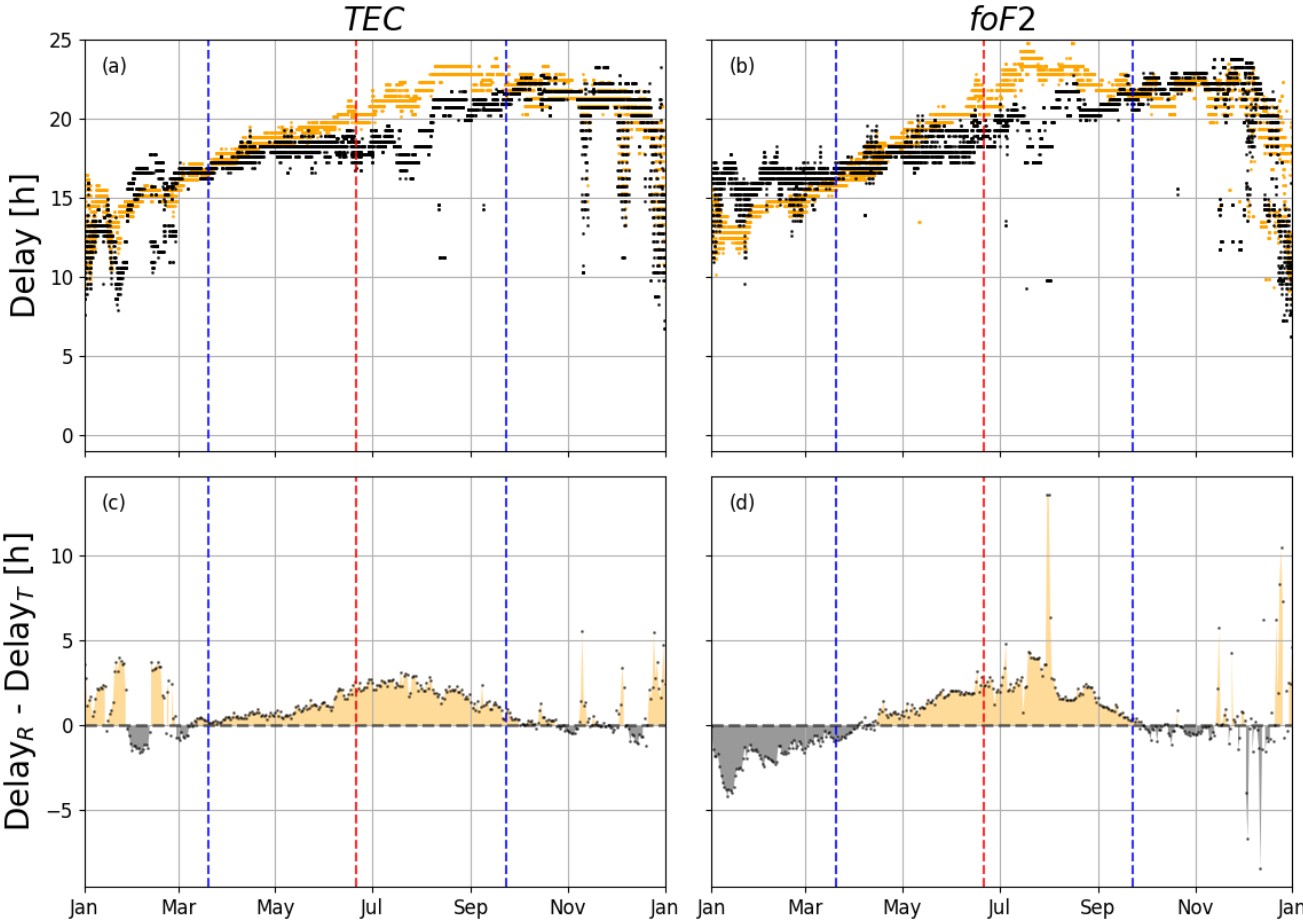

**Figure 13.** Superposed epoch plots for the delay (a, b) and difference in delays (c, d) for the ionospheric parameters TEC and foF2 with integrated EVE fluxes (6 to 105 nm) for Rome (orange) and Tromsø (black). The temporal resolution is one hour. Equinoxes are marked with the blue dashed lines and solstice is marked with the red dashed line.

delay decreases with increasing latitude. From $\approx 21.5$ hours at $30°N$ to $\approx 19.0$ hours at $70°N$, or $\approx -0.06$ hours per degree in latitude. Therefore, the delay maps confirm the latitudinal variations as seen in Figures 9 and 13. The variation in delay with longitude is small and does not show any dominant trend in winter. The variation of the delay with longitude in summer is much smaller than the variation in latitude for the same season, with a change of $\approx -0.01$ hours per degree in longitude. The small and similar magnetic declination for the European region could be related to the small variations of the ionospheric delay with longitude. There is an influence of the magnetic declination on the mid-latitude ionosphere, which leads to similar longitudinal transport processes in all seasons (Zhang et al., 2012, 2013). This behavior has to be explored with observational data for different regions or modeling efforts in the future.

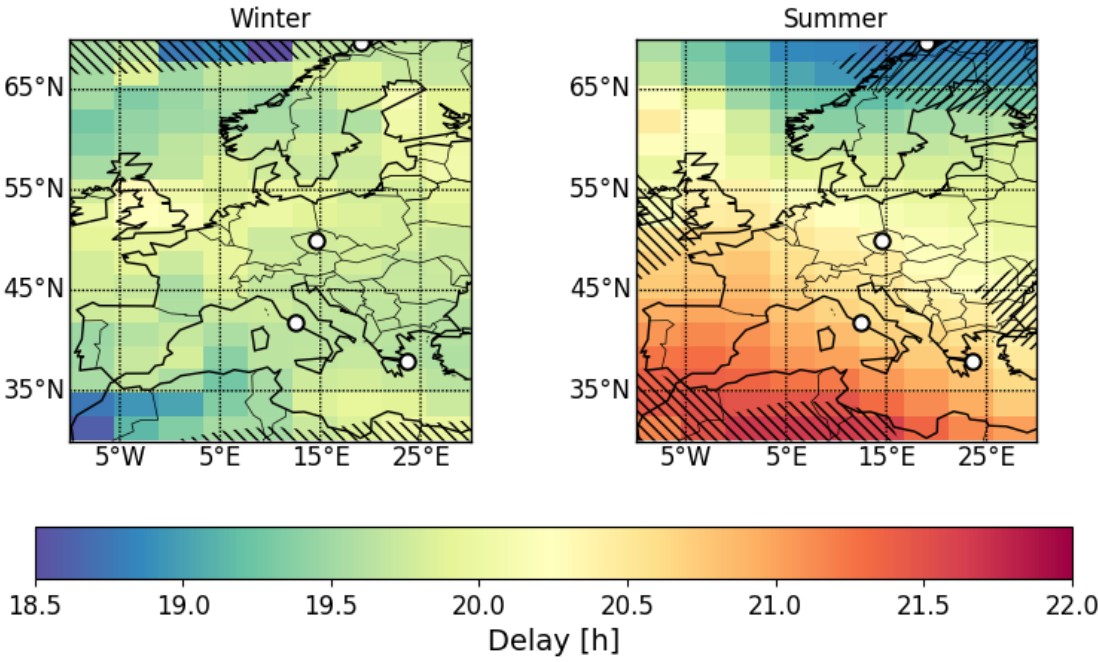

**Figure 14.** Map of the delay of TEC with respect to EUV in summer (May to August) and winter (November to February) within the time period from January 2011 to December 2013. The delay varies between $\approx 18.6$ and $\approx 21.7$ hours. The hatched regions on the map represent significantly greater (upper left to lower right fill) or smaller (upper right to lower left fill) correlations compared to the average of each map ($\pm$ one standard deviation). The absolute correlation coefficient is $\approx 0.28$ in summer and $\approx 0.17$ in winter. The ionosonde stations Tromsø, Průhonice, Rome and Athens are marked with the white dots.

The next analysis averages the calculated time series of delay maps over longitude to get a mean value for the delay at each latitude. The results are summarized with epoch plots in Figure 15, and have a resolution of one week (mean value) to allow a better presentation of the long-term changes of the ionospheric delay. The latitude-dependent time series in Figure 15 is consistent with the results and the assumed trend from the seasonal variations is present. In October, the delay reaches the same value for all latitudes and does not change any more until the sudden decrease in December, which happens for all latitudes.
The trend based on the geomagnetic activity (see Figures 4 and 5) is also represented in Figure 15.

## 6    Conclusions

The main challenge of delay calculation at high temporal resolution is the impact of the diurnal variations of ionospheric parameters. These have a impact on the calculated correlations coefficients, but do not influence the relative trend in a significant way. This study proved that a reliable delay calculation is possible at hourly resolution through different analysis: comparison

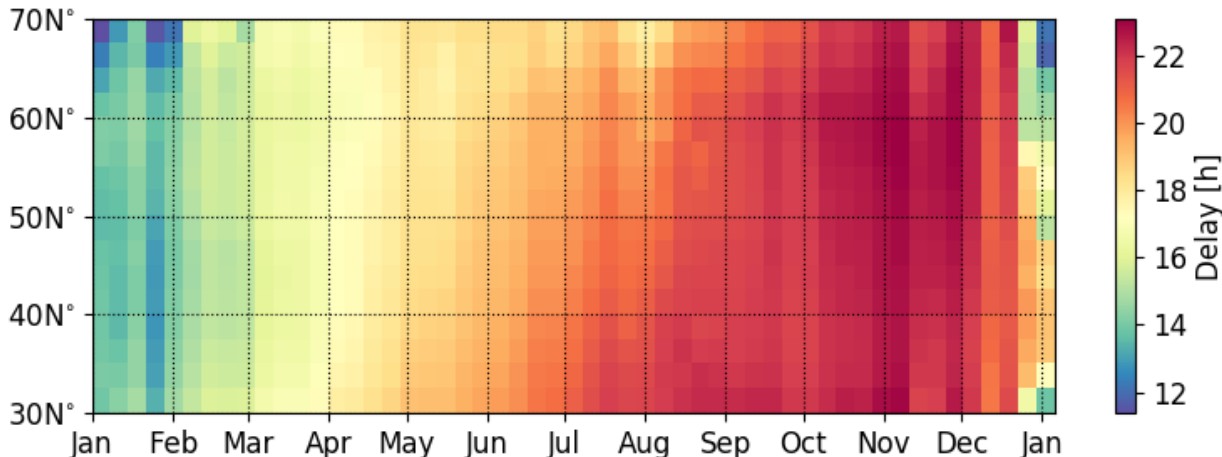

**Figure 15.** Time series of the delay of TEC with respect to EUV as an epoch plot for the mid-latitudes covering Europe within the time period from January 2011 to December 2013. The delay varies between $\approx 11.3$ and $\approx 23.1$ hours. The absolute correlation coefficient is $\approx 0.21$ during the period.

265 of delays between fixed local time, fixed location, and comparison of correlation coefficients on different sub-annual time scales. These results are important for future analysis of the delay at high temporal resolution.

The main analysis confirmed the findings of previous studies dealing with variations of the delayed ionospheric response to solar EUV with solar activity and latitude:

- The geomagnetic activity has a strong influence on the delay, which is visible as global trend in the delay within this
270  study. The strong impact of the geomagnetic activity was already suggested in other studies, e.g. Ren et al. (2018).

- The results indicate an influence of the 11-year solar cycle or at least an increase of the delay with increasing solar activity from year to year. This result is consistent with findings by Rich et al. (2003) and Chen et al. (2015).

The variability of the delayed ionospheric response to solar EUV with geomagnetic activity and the seasonal variations of the delay was shown with delay time series from January 2011 to December 2013. These findings allow the following conclusions:

275 - The comparison of the delay for locations in northern and southern hemisphere shows a seasonal variation, which occurs for both investigated ionospheric parameters TEC and foF2. The seasonal variation for foF2, which describes only the F2 layer, is larger compared with TEC of the whole ionosphere-plasmasphere system.

- The analysis of IGS TEC maps covering the European region indicates a latitudinal dependence of the delay for mid-latitudes, which is pronounced in summer and vanishes in winter. A North-South trend of the ionospheric delay during
280  summer month has been observed with $\approx 0.06$ hours per degree in latitude.

For the seasonal variation the difference in the delay was calculated at stations of similar latitude in both hemispheres for TEC with $\approx 5 \pm 0.7$ hours and foF2 with $\approx 8 \pm 0.8$ hours. The decrease of the delay with latitude in the European mid-latitudes from $\approx 21.5$ hours at 30°N to $\approx 19$ hours at 70°N in summer and the roughly constant delay of $\approx 19.5$ hours for the whole region in winter also show a seasonal difference in the delay.

Future analysis would benefit from high resolution ionospheric delay calculations for longer time periods that cover different solar and geomagnetic activity conditions. Such work will require ongoing efforts to measure the solar EUV radiation in the future, since these data are the basis for the delay calculations. The thermospheric conditions (e.g. neutral winds or composition changes in the atomic/molecular ratio), which are known for their impact on the ionosphere (Rishbeth, 1998; Rishbeth et al., 2000) should also be included in future analysis. Results presented in this study need to be further confirmed and studied by

model calculations. The underlying processes for the delayed ionospheric response to solar EUV radiation need to be described, since this knowledge presents an opportunity to validate or improve physics-based models.

*Data availability.* Kp-index data have been provided by NASA through https://omniweb.gsfc.nasa.gov/form/dx1.html. IGS TEC maps have been provided by NASA through ftp://cddis.gsfc.nasa.gov/gnss/products/ionex. Ionosonde data have been provided by NOAA ftp://ftp.ngdc.noaa.gov/ionosonde/. SDO-EVE data have been provided by Laboratory for

Atmospheric and Space Physics (LASP) through http://lasp.colorado.edu/eve/data_access/evewebdata. The shapefiles of the World Magnetic Model have been provided by NASA through ftp://ftp.ngdc.noaa.gov/geomag/wmm/wmm2015/shapefiles/. The International Geomagnetic Reference Field has been provided by NASA through https://www.ngdc.noaa.gov/IAGA/vmod/igrf.html.

*Author contributions.* E. Schmölter performed the calculations and composed the fist draft of the paper. J. Berdermann, N. Jakowski, and

Ch. Jacobi actively contributed to the analysis and paper writing.

*Competing interests.* Ch. Jacobi is one of the editors-in-chief of Annales Geophysicae.

*Acknowledgements.* IGS TEC maps, Kp-index data, International Geomagnetic Reference Field and shapefiles of the World Magnetic Model have been provided by NASA. EVE data has been provided by LASP. Ionosonde data has been provided by NOAA. The study has been supported by Deutsche Forschungsgemeinschaft (DFG) through grants No. BE 5789/2-1 and JA 836/33-1.

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
