# Peer review of "Spatial and seasonal effects on the delayed ionospheric response to solar EUV changes"

_Annales Geophysicae, 2019_

## Referee Comment (RC1) · Jiuhou Lei (Referee) · 9 Aug 2019

This manuscript mainly focused on the temporal and spatial variation of the delayed response of terrestrial ionosphere to the solar EUV flux changes that related to the solar rotation. This is an interesting study, and the paper is publishable in AG. However, there are some issues to be addressed before it is formally accepted for publication.

1. In Figure 3, the authors compared the correlation coefficients and the time delay retrieved from two datasets by fixing local time or fixing location. The authors should note that the time delay of ionosphere to the solar EUV flux change depends on solar local time. The time delay inferred from fixed location dataset can be partly considered as the averaged delay at different local times. The authors should point out this issue.

[Figure]

2. The time delay of ionospheric parameters is the key in this analysis. The difference between the time delay by using the 2 methods is greater than 4 hours. How about the uncertainty of the obtained time delay? In addition, the diurnal variation of ionospheric parameters may affect the calculation of time delay. They can provide the time delay by removing the diurnal variation in Figure 3. 3. Is the time delay reliable as the correlation coefficient is less than 0.4? 4. How do they calculate the Kp index, the red line in Figure 4? 5. In Figure 4, the Kp index, the correlation coefficient and time delay show similar decreases during the end of each year. The authors indicated that the lower correlation coefficient and time delay should be related to the corresponding lower geomagnetic activity. Why the correlation coefficient is lower when the geomagnetic disturbance (Kp) is lower? 6. In Figures 11 and 12, the time delay generally does not change with latitude in winter. Whereas, during winter time the correlation coefficient is nearly 0 as seen in Figure 4. Therefore, the absolute values of the correlation coefficient should be provided in Figures 11 and 12.

---

## Author Comment (AC1) · 15 Aug 2019

Thank you so much for the comments and suggestions to our manuscript! We will revise the manuscript according to your advice.

1. In Figure 3, the authors compared the correlation coefficients and the time delay retrieved from two datasets by fixing local time or fixing location. The authors should note that the time delay of ionosphere to the solar EUV flux change depends on solar local time. The time delay inferred from fixed location dataset can be partly considered as the averaged delay at different local times. The authors should point out this issue.

Answer: We will clarify the difference between local time and fixed location analysis as suggested.

[Figure]

"The delayed ionospheric response to solar EUV radiation depends on the solar local time and the calculated results for fixed locations can be understood as a mean ionospheric delay for different local times. The local time approach would be preferred for this reason. Nevertheless, the analysis with fixed local time is not used in the further analysis, since the extracted time series from the IGS TEC maps relies less on measurements considering areas with few or no ground stations."

2. The time delay of ionospheric parameters is the key in this analysis. The difference between the time delay by using the 2 methods is greater than 4 hours. How about the uncertainty of the obtained time delay? In addition, the diurnal variation of ionospheric parameters may affect the calculation of time delay. They can provide the time delay by removing the diurnal variation in Figure 3.

Answer: We will add an explanation of the mean difference (approximately 3.15 hours) between both approaches and characterized the uncertainty.

"The two different approaches have a mean variance of approximately 3.15 hours, which accounts for an uncertainty of approximately 16.04 % in the ionospheric delay calculation. This is an acceptable impact of the diurnal variation on the trend of the delay for characterizing temporal and spatial changes."

The time delay can be provided without the diurnal variation, but the available approaches don't improve the process or even have a negative impact. Removing the diurnal variation with a band-stop filter doesn't remove the diurnal variation completely and there is no improvement for the correlation and reliability of the delay (Schmölter et al., 2018). Calculating daily mean values for TEC or foF2 doesn't allow a delay analysis on hourly resolution, since that would require interpolation back to the high resolution and this in turn has a huge impact on the delay calculation. In general, an improvement of the correlation coefficients (e.g. calculating daily means) doesn't grant a more reliable or precise delay calculation. We decided against filters or changes on the signal, acknowledge the impact of the diurnal variation and focus on features of the

ionospheric delay, which are not defined by small scale changes. In addition, the calculated value range and features of the delay trend fit very well with results of preceding studies.

3. Is the time delay reliable as the correlation coefficient is less than 0.4?

Answer: The results are statistically significant due to the big sample size (90 days on hourly resolution) and, as shown by Figures 2 and 3, the relative trend of correlation coefficients and delay is not changed in different approaches. For example using fixed local times gives much higher correlation coefficients and the resulting annual variation of the delay is present.

We will clarify the reliability in the manuscript as mentioned in the reply to comment 2.

4. How do they calculate the Kp index, the red line in Figure 4?

Answer: We will clarify the description of Figure 4. The Kp-index data are shown in weekly resolution with the red line, because the trend on hourly or daily resolution doesn't give a meaningful overview for the long-term changes due to the much stronger short-term variations. A description for the calculation of the smoothed trends will be added as well.

"The transparent red lines or dots show the raw data: Kp-index in weekly resolution (a), correlation coefficients between EUV and TEC (b) and delays between EUV and TEC (c) in hourly resolution. The black lines show the smoothed weekly means to present the overall trend (running mean with window size of 10 days)."

5. In Figure 4, the Kp index, the correlation coefficient and time delay show similar decreases during the end of each year. The authors indicated that the lower correlation coefficient and time delay should be related to the corresponding lower geomagnetic activity. Why the correlation coefficient is lower when the geomagnetic disturbance (Kp) is lower?

Answer: The explanation of annual variations of the correlation coefficients and ionospheric delay with geomagnetic activity is difficult and requires modeling efforts in the future. The topic introduces a lot of complexity due to the various ways geomagnetic activity impacts the ionospheric state. An explanation could be the global F2 layer ionization due to geomagnetic activity (Lal, C. ( 1992), Global F2 layer ionization and geomagnetic activity, J. Geophys. Res., 97( A8), 12153– 12159, doi:10.1029/92JA00325.). As already suggested in the manuscript, analyses for longer time periods are required to further explain this relation and the processes behind it.

6. In Figures 11 and 12, the time delay generally does not change with latitude in winter. Whereas, during winter time the correlation coefficient is nearly 0 as seen in Figure 4. Therefore, the absolute values of the correlation coefficient should be provided in Figures 11 and 12.

Answer: We will add the absolute correlation coefficients to Figures 11 and 12 and modify the captions accordingly.

"Figure 11: Map of the delay of TEC with respect to EUV in summer (May to August) and winter (November to February) within the time period from 2011 to 2014. The delay varies between ≈18.6 and ≈21.7 hours. The hatched regions on the map represent significantly greater (upper left to lower right fill) or smaller (upper right to lower left fill) correlations compared to the average of each map (± one standard deviation). The absolute correlation coefficient is ≈0.28 in summer and ≈0.17 in winter."

"Figure 12: Time series of the delay of TEC with respect to EUV as an epoch plot for the mid-latitudes covering Europe within the time period from 2011 to 2013. The delay varies between ≈11.3 and ≈23.1 hours. The absolute correlation coefficient is ≈0.21 during the period."

[Figure]

---

## Referee Comment (RC2) · Anonymous Referee #2 · 19 Aug 2019

The article **Spatial and seasonal effects on the delayed ionospheric response to solar EUV changes** presents a study that examines the relationship between solar EUV irradiance and F-region ionospheric density. This study builds off of previous work, confirming previous results using higher resolution calculations. It also examines seasonal and latitude variations for a small region of the globe. Both the validation efforts and the study into local European variations are of interest to the scientific community. The presentation and language are not high enough quality for publication and the authors do not consistently give proper credit to related work. The length of the paper is adequate. I believe this study could contribute positively to the scientific community if substantial changes are made.

[Figure]

**1 Title and abstract**

The title is clear and appropriate. Although it contains an acronym, the acronym is commonly known throughout the community and does not have common alternate definitions. Grammatical corrections include:

1. (Line 1) "...EUV radiation to analyse the delayed ionospheric response to test and improve previous studies on the ionospheric delay. Several..."

2. (Line 4) " ..the analysis at an hourly resolution..."

3. (Line 13) "...Results confirm that geomagnetic activity and the 11-year solar cycle also affect the ionospheric response to solar EUV changes"

Alternatively, lines 6–14 could be re-written to more accurately summarise the conclusions.

**2 Major Issues and Questions**

1. The motivation provided through GNNS in the introduction (around Line 25) is not appropriate. If the authors wish to continue with this motivation, the following issues need to be addressed:

   (a) Not all terms are defined (e.g., a different definition of "high temporal resolution" is used on line 27 when compared to the rest of the paper)

   (b) Citations to GNSS work are absent. The motivation would be strengthened by citations of articles that have proved high accuracy GNSS products require accurate ionospheric models, as well as citations to articles that highlight missing physics in ionospheric models when handling the ionospheric

delay. Given studies such as Ren et al. (2018), which show that ionospheric models do capture the ionospheric delay to solar EUV irradiance, I would recommend that the authors find a different motivation for their study.

(c) Finally, this motivation also requires citations that demonstrate that other higher order GNSS correction issues (such as the bending terms) are not as important as the parts of the refractive index terms that would be affected by the (to have been) demonstrated issues with ionospheric models that are affected by the ionospheric delay.

2. The authors highlight the differences between the regions covered by the two ionospheric parameters used in this study. While it is true that GNSS TEC includes information about the entire ionosphere-plasmasphere system through which it travels, it is also true that the $F_2$ region is responsible for most of variations in TEC (e.g., Petrie et al. 2011). Text and data interpretations would benefit from clarifying the relative contributions from the different ionospheric regions and plasmasphere to the TEC, as well as the expected agreement between the column integrated plasma density and the critical frequency of the $F_2$ layer based on past studies.

3. In the introduction, the authors do not sufficiently discuss the contributions of previous ionospheric delay studies. Specifically, there is no discussion as to the physical reason behind the ionospheric delay, although this has previously been investigated (e.g., Ren et al. 2018).

4. A motivation behind using the European and Australian regions is needed. For example, why not use North and South America (see coverage for 1 January 2011 in the attached Figure)? This figure is included not to say that there is not a good reason to use European and Australian data, but to show that "good data coverage for Europe" is not a good reason in and of itself.

5. The authors state that they use two important ionospheric parameters that are appropriate to investigate the processes responsible for the ionospheric delay (data section), but they state in the conclusions that the processes for the delayed ionospheric response still need to be described. If the first statement is true, then an investigation of the underlying physical processes should be included in this paper. If such a study is beyond the scope of this paper, than the statements made about the ionospheric parameters used to study the characteristics of the ionospheric delay should be altered.

6. The authors state that the TEC is more important than the fo$F_2$ but do not back up this valuation, especially since they say in the introduction that the ionospheric delay for the two parameters is very similar. The reason given, "TEC is...less sensitive to disturbances, such as plasma redistribution, than other parameters" is not substantiated. Additionally, since TEC is regularly used to study plasma redistribution (e.g., Foster 2008; doi:10.1029/181GM12) , the degree of sensitivity difference between TEC and foF$_2$ needs to be shown to be significant (either by the authors or through appropriate referencing) for this valuation to be believable.

7. (Line 64) The resampling method needs to be described in more detail. Was an interpolation used? If so, between which points? Was the nearest value taken?

8. (Line 67) What is the temporal resolution of the ionosonde data and were they hand scaled or autoscaled?

9. What is the effect of the difference in geographic longitude and magnetic location (including location relative to the auroral oval, declination, and inclination) on the locations in Europe and Australia?

10. (Line 76) How are data resampled in this instance? From the context, it appears that the authors are downsampling data from a minute-scale resolution to a one

hour resolution, but this is unclear (especially since the same wording was used for a different process on line 64).

11. (Line 90) A better explanation of why the correlation coefficient is still useful even though the values specify that the data sets being compared are uncorrelated is needed.

12. What data quality constraints were applied to the input and processed data? Why were periods when the data quality is stated to be poor included?

13. The authors state that the ionospheric delays show a good correlation with the geomagnetic activity, but this is not demonstrated. If the authors believe that they have demonstrated this correlation, they should improve the clarity of the figure presentation and the text surrounding it.

14. (Line 184) The authors are quick to attribute differences between the TEC and fo$F_2$ ionospheric delays to differences between the $F_2$ peak and the ionosphere-plasmasphere system, but there are other possibilities (including the background model used in the TEC calculation) that should be acknowledged or eliminated.

15. There appears to be an offset between solstice and equinox occurrence and the seasonal variations shown in (Figure 10). Why is this? Has it been seen before?

16. Figures 7 and 8 show a lot of scatter at the individual stations. The analysis presented in section 5 makes claims about latitude variations based on these figures that do not appear to be significant, due to this scatter. This analysis would be improved by including another figure with delay differences between the sites or, possibly, by adding confidence bars (perhaps standard deviations) to the hourly delays in Figures 7 and 8.

17. (Line 191) The authors state that the latitudinal dependence in the European sector is not visible in the winter. However, Figure 7b shows a latitudinal variation

that is perhaps clearer than that in the summer, just different.

18. How is the resampling for Figure 12 performed? Is a running period or binned week used? Clarify this analysis process so that others may reliably replicate these results.

19. (Line 211) What about the winter variations? Does the longitudinal ionospheric delay variation have a seasonal variation at all? It seems likely that this lack of variation is related to the small range of magnetic declination over Europe, which leads to longitudinally similar ionospheric transport processes regardless of season. Whatever the authors believe the reason to be, it should be discussed.

20. The last sentence of the conclusions omits the work done by Ren et al. (2018). The article would be improved by a discussion of the results in the context of the physical mechanism presented in that article and also by providing a clearer motivation behind using the ionospheric delay to validate or improve physics-based models.

**3   Figures and tables**

1. (Table 1 caption): "...provide an approximate ionospheric delay to solar activity at a daily resolution."

2. (Figure 1): This figure would benefit by over-plotting magnetic field information (such as the IGRF declination or at the $hmF_2$) and the geomagnetic equator.

3. (Table 2): Which magnetic coordinate system is used for the geomagnetic coordinates?

4. (Figure 2): Rows should be labeled with "Weekly", "Daily", and "Hourly"

5. (Figure 2 caption): "...data, as well as the resulting correlation coefficients (red), for..."

6. (Figure 10): Mark the locations of the equinoxes and solstices.

7. (Figure 11): Mark the locations of the European stations, to improve comparisons between Figure 11 and Figure 7.

**4 Grammar and organisation**

1. Throughout the paper both $3^{rd}$ person and impersonal tenses are used. This should be changed so that the tense throughout the article is consistent

2. Throughout the paper approximations are used for numbers that do not need them (e.g., the locations on Line 95 specify the approximate location of Rome and this is already appropriately expressed by limiting the number of significant figures)

3. (Lines 27, ) "which" should either be preceded by a comma or replaced with "that"

4. (Line 17) "dominating" should be "dominant"

5. (Line 20) "...ionospheric variations that may depend on time or location."

6. (Line 21) "...in the solar spectrum'...'. This change is necessary because the authors, in this sentence, are referring to the entire ionosphere, which means that X-rays and higher energy irradiance that impact the D and E regions are also important.

7. (Line 22) "...and composition at specific..."

8. (Line 23) "...electron density distribution. An understanding of the ionospheric chemical and physical processes is important, since..."

9. (Line 28) remove duplicated text "is needed"

10. (Line 32) "...have revealed that ionospheric parameters have a delayed response to solar variability. A selection of these studies..."

11. (Line 33) "...was calculated using different EUV proxies or measurements of the EUV flux at daily resolutions."

12. (Line 35) "...the delay at a higher temporal resolution of one hour. Furthermore, we examine the hemispheric dependence of the ionospheric delay with a detailed study of the European region."

13. (Line 37) "is made based on" should be "uses"

14. (Line 37) "The" needed before "Time series"

15. (Line 43) "...the ionosphere without complicating contributions from the plasmasphere and lower ionospheric layers. As expected, the results..."

16. (Lines 45-47) This text belongs in the data or analysis section, not the introduction.

17. (Data) This section would benefit by subsections for either the different data sources or between the presentation of the data sources and the data analysis techniques

18. (Line 49) "...spectrum have been continuously measured since 2000 C.E., with EUV observational data publicly available from..."

19. (Line 55) "...have a temporal resolution of 20 seconds. EVE data also cover several years (2011 to 2014)..."

20. (Line 106) Description of the IGS TEC maps belongs in the Data section.

21. (Line 108) remove comma between "show" and "that"

22. (Line 109) "...be calculated at an hourly resolution for fixed..."

23. (Line 123) The sentence, "Se do not see any...different variations" is confusing and should be rewritten.

24. (Line 124) "...keeping in mind that their magnitude may differ due to..."

25. (Line 128) "...in Table 1. For example, Jakowski et al. (1991) used the..."

26. (Line 129) "...satellite-based EUV-TEC measurements (Unglaub et al., 2011) and also calculated the delay with EUV fluxes. The validation with EVE EUV flux measurements was important because the solar rotation variations..."

27. (Line 135) The first two sentences of this paragraph belong in the introduction. The remaining sentences belongs in the data section.

28. (Line 138) Which calculation are the authors referring to?

29. (Line 144) "...negative values. In Figure 4, this was interpreted as a possible effect of geomagnetic activity."

30. (Line 145) "...time period, the correlation coefficient drops due to data gaps and the applied interpolation method. (start new paragraph after this sentence)"

31. (Line 146) "...are smaller than those of the TEC. However, the trends of the two correlation coefficients are similar for the..."

32. (Line 148) "...Tromsø again show that the largest deviation from..."

33. (Line 152) "The TEC and foF$_2$ correlation coefficients for the Australian stations are shown in Figure 6. In general, the Australian correlation..."

34. (Line 155) "...these results. Most notably, the decrease and..."

35. (Line 156) Which seasonal variations do the authors expect to be impactful? Referencing is appropriate but the text description should be slightly more detailed.

36. (Line 174) "...the delay at a daily resolution for longer time periods than the one used in this study."

37. (Line 175) The sentence is unclear and needs to be reworded.

38. (Line 179) What do the authors mean by "global trend"?

39. (Line 179) The sentence is unclear and needs to be reworded.

40. (Line 183) "...a stronger seasonal variation..."

41. (Line188) "...with latitude in northern summer. The station at..."

42. (Line 193) remove "where data from high latitudes are missing" because the Australian stations have a larger low-latitude extent than the European stations and this phrase does not reflect that.

43. (Line 194) recommend replacing "agree" with "are consistent"

44. (Line 197) "...good observational coverage..."

45. (Line 198) Remove repeated description of the IGS TEC map.

46. (Line 202) "...Figure 11, which maps the mean delay values for the mid-latitudes in summer (May-August) and winter (November-February). Figure 11 shows delays that are consistent with the results from the European ionosonde stations (Figure 7b)."

47. (Line 205) "...hours over the entire region."

48. (Line 205) The sentence that begins at the end of this line is unclear and needs to be reworded.

49. (Line 207) "...the delay decreases with increasing latitude. From..."

50. (Line 208) "...70°N, or about -0.06 hours per degree in latitude."

51. (Line 211) "...is much smaller than the variation in latitude for the same season, with a change of..."

52. (Line 224) "...main analysis, we confirmed..."

53. (Line 234) Move the last two bullet points starting on this line to the previous paragraph where the authors were discussing the portions of previous studies that this study validated.

54. (Line 242) "Future analysis would benefit from high resolution ionospheric delay calculations for longer time periods that cover different..."

55. (Line 243) Sentence starting at the end of this line is unclear and needs to be reworded.

56. (Data availability and acknowledgements) Not all acronyms are defined.

57. (Line 297) "F 2" should be "$F_2$"

58. (Line 327) page numbers missing and filled using n/a–n/a

**5 Referencing**

1. (Line 22) Reference needed for the impact of solar irradiance on the vertical ionospheric structure

2. (Line 44) Citation to a source that explains or demonstrated that TEC is dominated by the $F_2$ peak response is needed

3. (Line 63) Which model is included in the TEC calculation? Include a very short description and a citation to this model.

4. (Line 106) Citation for IGS TEC maps needed.

5. (Line 196) Citation needed.

[Figure]

**Fig. 1.**

---

## Author Comment (AC2) · 2 Sep 2019

Thank you so much for the detailed comments and suggestions to our manuscript! We will revise the manuscript according to your advice.

1. Title and abstract (Line 1) "...EUV radiation to analyse the delayed ionospheric response to test and improve previous studies on the ionospheric delay. Several..."

(Line 4) " ..the analysis at an hourly resolution..."

(Line 13) "...Results confirm that geomagnetic activity and the 11-year solar cycle also affect the ionospheric response to solar EUV changes"

[Figure]

Alternatively, lines 6–14 could be re-written to more accurately summarise the conclusions.

Answer: We will change the abstract as suggested.

2. Major Issues and Questions

The motivation provided through GNNS in the introduction (around Line 25) is not appropriate. If the authors wish to continue with this motivation, the following issues need to be addressed:

(a) Not all terms are defined (e.g., a different definition of "high temporal resolution" is used on line 27 when compared to the rest of the paper).

(b) Citations to GNSS work are absent. The motivation would be strengthened by citations of articles that have proved high accuracy GNSS products require accurate ionospheric models, as well as citations to articles that highlight missing physics in ionospheric models when handling the ionospheric delay. Given studies such as Ren et al. (2018), which show that ionosphericmodels do capture the ionospheric delay to solar EUV irradiance, I would recommend that the authors find a different motivation for their study.

(c) Finally, this motivation also requires citations that demonstrate that other higher order GNSS correction issues (such as the bending terms) are not as important as the parts of the refractive index terms that would be affected by the (to have been) demonstrated issues with ionospheric models that are affected by the ionospheric delay.

Answer: We will change the motivation for our study:

"The delayed ionospheric response to solar EUV radiation is captured in various ionospheric models (Ren et al., 2018; Vaishnav et al., 2018) and respective simulations can confirm results of previous studies estimating the ionospheric delay with observational data on daily resolution. The calculation of the delay with observational data in high temporal resolution ($\leq 1$ hour) is of interest to describe features like seasonal and spatial variations in more detail. The dependence on solar and geomagnetic activity (Ren et al., 2018) can be explored further. In the future, results for the ionospheric delay on high temporal resolution will strengthen the understanding of ionospheric processes and help to validate physics-based models."

The authors highlight the differences between the regions covered by the two ionospheric parameters used in this study. While it is true that GNSS TEC includes information about the entire ionosphere-plasmasphere system through which it travels, it is also true that the F2 region is responsible for most of variations in TEC (e.g., Petrie et al. 2011). Text and data interpretations would benefit from clarifying the relative contributions from the different ionospheric regions and plasmasphere to the TEC, as well as the expected agreement between the column integrated plasma density and the critical frequency of the F2 layer based on past studies.

Answer: We will add an explanation about the ionospheric and plasmaspheric contribution to TEC and clarify the dominant role of the F2 layer:

"The variations of TEC are mostly controlled by the F2 layer (Lunt et al., 1999; Petrie et al., 2010; Klimenko et al., 2015) and for mid-latitudes the total plasmaspheric contribution to TEC is between approximately 8 to 15 % during daytime and approximately 30 % during nighttime (Yizengaw et al., 2008)."

We will add a clarification for the high correlation between TEC and foF2, but also mention the difference of both parameters, which could result in different ionospheric delays.

"Both ionospheric parameters are highly correlated (Kouris et al., 2004), but variations like different peak time of the diurnal variation (Liu et al., 2014) could have a considerable impact on the delayed ionospheric response."

In the introduction, the authors do not sufficiently discuss the contributions of previous ionospheric delay studies. Specifically, there is no discussion as to the physical reason
behind the ionospheric delay, although this has previously been investigated (e.g., Ren et al. 2018).

We will add a summary of the recent investigations by Ren et al. 2018 to give an overview to the processes behind the delay:

"The recent results by Ren et al. 2018 from observational and model calculations specified different features of the ionospheric delay. A strong impact of the geomagnetic activity on the ionospheric delay to solar EUV changes was found. Simulations with the Thermosphere Ionosphere Electrodynamics General Circulation Model (TIEGCM) and calculations were used to discuss the influence of ion production and loss as well as the impact of the O/N2 ratio. The ion production responds immediately to EUV variations and depends on both, the solar EUV flux contribution and the O/N2 ratio. The loss is delayed and controlled by the O/N2 ratio, which in turn is also dominated by the solar EUV flux contribution. The ionospheric response could further be modulated by dynamic and electrodynamic processes in the ionosphere. Ren et al. 2018 also showed a latitudinal dependence of the delay."

A motivation behind using the European and Australian regions is needed. For example, why not use North and South America (see coverage for 1 January 2011 in the attached Figure)? This figure is included not to say that there is not a good reason to use European and Australian data, but to show that "good data coverage for Europe" is not a good reason in and of itself.

Answer: We will add a reference to back our statement about data quantity/quality of TEC and ionosonde data for the European region:

"The dense coverage of GPS stations over Europe allows a good comparison with TEC data for these locations (Belehaki et al., 2015)."

We will clarify that not only the European region has good data coverage:

"The availability of TEC in maps with good data coverage for certain regions (e.g. European or North American region) allows a spatial analysis of the delay and a comparison with the foF2 data for specific locations."

We will add magnetic declination and inclination in Table 2 and explain further, why a comparison between both regions seems appropriate:

"The conditions of Earth's magnetic field for the European and Australian stations are comparable with small magnetic declination and similar absolute value of magnetic inclination (see Table 2). The selected stations seem appropriate for a comparison between northern and southern hemisphere due to these similar conditions."

The authors state that they use two important ionospheric parameters that are appropriate to investigate the processes responsible for the ionospheric delay (data section), but they state in the conclusions that the processes for the delayed ionospheric response still need to be described. If the first statement is true, then an investigation of the underlying physical processes should be included in this paper. If such a study is beyond the scope of this paper, than the statements made about the ionospheric parameters used to study the characteristics of the ionospheric delay should be altered.

Answer: We will change the statement to clarify the actual use of the parameters in the study:

"In the analysis, we correlate EUV with two important ionospheric parameters, appropriate to investigate features of the ionospheric delay."

The authors state that the TEC is more important than the foF2 but do not back up this valuation, especially since they say in the introduction that the ionospheric delay for the two parameters is very similar. The reason given, "TEC is...less sensitive to disturbances, such as plasma redistribution, than other parameters" is not substantiated. Additionally, since TEC is regularly used to study plasma redistribution (e.g., Foster 2008; doi:10.1029/181GM12) , the degree of sensitivity difference between TEC and foF2 needs to be shown to be significant (either by the authors or through appropriate referencing) for this valuation to be believable.

Answer: We will change the statement to clarify, why TEC is used:

"The first parameter is TEC, which is an integral measurement of the electron density and well suited for the analysis of the ionospheric response to solar EUV variations. The parameter was used in several preceding studies to calculate the ionospheric delay (see Table 1)."

(Line 64) The resampling method needs to be described in more detail. Was an interpolation used? If so, between which points? Was the nearest value taken?

Answer: We will clarify the process: the data were extracted from IGS TEC maps without any interpolation (spatial or temporal).

"In preparation for the delay calculation, TEC values at seven ionosonde locations and one region (Europe) were extracted from the IGS TEC maps. For each ionosonde location the nearest grid point in the maps was used."

(Line 67) What is the temporal resolution of the ionosonde data and were they hand scaled or autoscaled?

Answer: We will clarify the temporal resolution and scaling of the ionosonde data:

"The other ionospheric parameter included in the analysis, foF2, is derived from ionosonde station data (NOAA, 2019) provided by the National Oceanic and Atmospheric Administration (NOAA), and are available for the same time periods with temporal resolution of 15 minutes (Wright and Paul, 1981)."

"In the northern hemisphere, the European stations Tromsø, Pruhonice, Rome, and Athens were selected (auto scaled), [. . .]"

"Instead we use data from the Australian stations Darwin, Camden, and Canberra for the analysis in the southern hemisphere (auto scaled)."

What is the effect of the difference in geographic longitude and magnetic location (including location relative to the auroral oval, declination, and inclination) on the locations in Europe and Australia?

Answer: We will add the declination and inclination of each location to Table 1 showing again the similar conditions for the comparison between northern and southern hemisphere. The specific conditions for Tromso (the only auroral station) are already explained in the manuscript.

(Line 76) How are data resampled in this instance? From the context, it appears that the authors are downsampling data from a minute-scale resolution to a one hour resolution, but this is unclear (especially since the same wording was used for a different process on line 64).

Answer: The wording for the method in line 64 was adjusted (see comment 7). We will change the description to clarify the use of the mean value to calculate the resampled data sets:

"In preparation of the analysis, all data are resampled to an hourly resolution using the mean value [. . .]"

(Line 90) A better explanation of why the correlation coefficient is still useful even though the values specify that the data sets being compared are uncorrelated is needed.

Answer: We will add an explanation, why the analysis of times with high and low correlation coefficients between solar EUV and ionospheric parameters is useful/important:

"The varying correlation between solar EUV flux or solar proxies like F10.7 with TEC is known from preceding studies. Solar EUV radiation is not able to describe the ionospheric variability at all time periods and on all time scales sufficient resulting in low correlation coefficients (Unglaub et al., 2012) and indicating a stronger impact of other processes (Verkhoglyadova et al., 2013). Analyzing both, times of high and low correlation between solar EUV flux and ionospheric parameters, is important to understand the changes of processes and interactions in the ionosphere on the whole."

What data quality constraints were applied to the input and processed data? Why were periods when the data quality is stated to be poor included?

Answer: We didn't accept data into our analysis with data gaps for longer time periods (e.g. gaps of several days) or with periods with lots of smaller data gaps. The correlation coefficients were then calculated for the whole available time period to get an impression, how periods of poor data quality impact the results. In Figure 7 and 8 these periods are removed due to the uncertainty for the delay calculation.

The authors state that the ionospheric delays show a good correlation with the geomagnetic activity, but this is not demonstrated. If the authors believe that they have demonstrated this correlation, they should improve the clarity of the figure presentation and the text surrounding it.

Answer: We will add a Figure comparing the gradient of the delay with the Kp-index and explain the modulation of the geomagnetic activity on the delay. The correlation coefficients in each year are 0.53 in 2011, 0.70 in 2012 and 0.77 in 2013 (see supplements).

(Line 184) The authors are quick to attribute differences between the TEC and foF2 ionospheric delays to differences between the F2 peak and the ionosphere-plasmasphere system, but there are other possibilities (including the background model used in the TEC calculation) that should be acknowledged or eliminated.

Answer: We will clarify the concerns about other impacts causing the difference in TEC and foF2 results:

"These results could indicate a strong seasonal variation of the ionospheric delay in the F2 layer compared to the whole ionosphere-plasmasphere system, but there are other possible sources for the difference (e.g. the background model of the IGS TEC

maps)."

There appears to be an offset between solstice and equinox occurrence and the seasonal variations shown in (Figure 10). Why is this? Has it been seen before?

Answer: We will mark solstice and equinoxes in the Figure. An offset for the difference in the delays doesn't appear consistently. In addition, such a detailed analysis of features in the seasonal variations requires model calculations to eliminate uncertainties due to the specific locations. This question could be addressed in future studies though.

Figures 7 and 8 show a lot of scatter at the individual stations. The analysis presented in section 5 makes claims about latitude variations based on these figures that do not appear to be significant, due to this scatter. This analysis would be improved by including another figure with delay differences between the sites or, possibly, by adding confidence bars (perhaps standard deviations) to the hourly delays in Figures 7 and 8.

Answer: We will add a figure similar to the comparison between Rome and Canberra for Rome and Tromso (see supplement) to show the difference between the two stations and discuss the trend of the difference with latitude in more detail (especially in regards of the different variation with latitude in winter mentioned in comment 17):

"Figure 12 shows the difference between the stations Rome and Tromso for both ionospheric parameters. The results for TEC show a greater or similar ionospheric delay for the station Rome compared to the station Tromso. There are only a few short time periods during winter with a greater ionospheric delay for the station Tromso. A stronger seasonal variation appears for the parameter foF2, but overall the ionospheric delay is still greater for the station Rome. The mean difference for results in Figure 12 is 1.08 hours for TEC and 0.52 hours for foF2. The changes with latitudinal dependence of the trends during winter are due to the stronger increase of the ionospheric delay for Rome during summer."
(Line 191) The authors state that the latitudinal dependence in the European sector is not visible in the winter. However, Figure 7b shows a latitudinal variation that is perhaps clearer than that in the summer, just different.

Answer: We will clarify, that there is a difference in winter (see comment 16).

How is the resampling for Figure 12 performed? Is a running period or binned week used? Clarify this analysis process so that others may reliably replicate these results.

Answer: We will clarify that the weekly value is calculated with the mean:

"The results are summarized with epoch plots in Figure 12 having a resolution of one week (mean value) to allow a better presentation of the long-term changes of the ionospheric delay."

(Line 211) What about the winter variations? Does the longitudinal ionospheric delay variation have a seasonal variation at all? It seems likely that this lack of variation is related to the small range of magnetic declination over Europe, which leads to longitudinally similar ionospheric transport processes regardless of season. Whatever the authors believe the reason to be, it should be discussed.

Answer: We will discuss the possible explanation of the lack of longitudinal variations with the similar declination for the whole European region:

"The small and similar magnetic declination for the European region could be related to the small variations of the ionospheric delay with longitude. There is an influence of the magnetic declination on the mid-latitude ionosphere, which leads to similar longitudinal transport processes in all seasons (Zhan et al., 2012, 2013). This behavior has to be explored with observational data for different regions or modeling efforts in the future."

The last sentence of the conclusions omits the work done by Ren et al. (2018). The article would be improved by a discussion of the results in the context of the physical mechanism presented in that article and also by providing a clearer motivation behind using the ionospheric delay to validate or improve physics-based models.

Answer: We will change our motivation and add an explanation of the work by Ren et al. (2018) in our introduction (see comment 1 and 3).

3. Figures and tables (Table 1 caption): "...provide an approximate ionospheric delay to solar activity at a daily resolution.

Answer: We will change the description according to the suggestion.

(Figure 1): This figure would benefit by over-plotting magnetic field information (such as the IGRF declination or at the hmF2) and the geomagnetic equator.

Answer: We will plot the magnetic field information in the figure with data from the World Magnetic Model provided by NASA.

(Table 2): Which magnetic coordinate system is used for the geomagnetic coordinates?

Answer: We will add magnetic declination and inclination to the Table. All magnetic parameters will be calculated with the IGRF.

(Figure 2): Rows should be labeled with "Weekly", "Daily", and "Hourly"

Answer: We will add the labels to the figure.

(Figure 2 caption): "...data, as well as the resulting correlation coefficients (red), for..."

Answer: We will change the caption.

(Figure 10): Mark the locations of the equinoxes and solstices.

Answer: We will mark equinoxes and solstices and add a discussion (see major comment 15).

(Figure 11): Mark the locations of the European stations, to improve comparisons between Figure 11 and Figure 7.
Answer: We will add marks for the locations of the European stations.

4. Grammar and organisation

Throughout the paper both 3rd person and impersonal tenses are used. This should be changed so that the tense throughout the article is consistent

Answer: We will remove the use of the 3rd person.

Throughout the paper approximations are used for numbers that do not need them (e.g., the locations on Line 95 specify the approximate location of Rome and this is already appropriately expressed by limiting the number of significant figures)

Answer: We will remove the use of approximations for numbers that don't need them.

(Lines 27, ) "which" should either be preceded by a comma or replaced with "that" 4 (Line 17) "dominating" should be "dominant 5 (Line 20) "...ionospheric variations that may depend on time or location." 6 (Line 21) "...in the solar spectrum'...'. This change is necessary because the authors, in this sentence, are referring to the entire ionosphere, which means that X-rays and higher energy irradiance that impact the D and E regions are also important. 7 (Line 22) "...and composition at specific..." 8 (Line 23) "...electron density distribution. An understanding of the ionospheric chemical and physical processes is important, since..." 9 (Line 28) remove duplicated text "is needed" 10 (Line 32) "...have revealed that ionospheric parameters have a delayed response to solar variability. A selection of these studies..." 11 (Line 33) "...was calculated using different EUV proxies or measurements of the EUV flux at daily resolutions." 12 (Line 35) "...the delay at a higher temporal resolution of one hour. Furthermore, we examine the hemispheric dependence of the ionospheric delay with a detailed study of the European region." 13 (Line 37) "is made based on" should be "uses" 14 (Line 37) "The" needed before "Time series" 15 (Line 43) "...the ionosphere without complicating contributions from the plasmasphere and lower ionospheric layers. As expected, the results... 16 (Lines 45-47) This text belongs in the data or analysis section, not the introduction.

Answer: We will change the manuscript according to the suggestions.

(Data) This section would benefit by subsections for either the different data sources or between the presentation of the data sources and the data analysis techniques

Answer: We will add two subsections: "Solar EUV radiation" and "Ionospheric parameters".

(Line 49) "...spectrum have been continuously measured since 2000 C.E., with EUV observational data publicly available from..." 19 (Line 55) "...have a temporal resolution of 20 seconds. EVE data also cover several years (2011 to 2014)... 20 (Line 106) Description of the IGS TEC maps belongs in the Data section. 21 (Line 108) remove comma between "show" and "that" 22 (Line 109) "...be calculated at an hourly resolution for fixed..." 23 (Line 123) The sentence, "Se do not see any...different variations" is confusing and should be rewritten. 24 (Line 124) "...keeping in mind that their magnitude may differ due to..." 25 (Line 128) "...in Table 1. For example, Jakowski et al. (1991) used the..." 26 (Line 129) "...satellite-based EUV-TEC measurements (Unglaub et al., 2011) and also calculated the delay with EUV fluxes. The validation with EVE EUV flux measurements was important because the solar rotation variations..." 27 (Line 135) The first two sentences of this paragraph belong in the introduction. The remaining sentences belongs in the data section.

Answer: We will change the manuscript according to the suggestions.

(Line 138) Which calculation are the authors referring to?

Answer: We will clarify that the calculation of the ionospheric delay is referred to.

(Line 144) "...negative values. In Figure 4, this was interpreted as a possible effect of geomagnetic activity." 30 (Line 145) "...time period, the correlation coefficient drops due to data gaps and the applied interpolation method. (start new paragraph after this sentence)" 31 (Line 146) "...are smaller than those of the TEC. However, the trends of the two correlation coefficients are similar for the..." 32 (Line 148) "...Tromsø again

show that the largest deviation from..." 33 (Line 152) "The TEC and foF2 correlation coefficients for the Australian stations are shown in Figure 6. In general, the Australian correlation..." 34 (Line 155) "...these results. Most notably, the decrease and..."

Answer: We will change the manuscript according to the suggestions.

(Line 156) Which seasonal variations do the authors expect to be impactful? Referencing is appropriate but the text description should be slightly more detailed.

Answer: We will clarify which thermospheric conditions and season variations are meant:

"The difference might be due to further impacts on the correlation, e.g. thermospheric wind conditions or seasonal variations due to composition changes (atomic/molecular ratio), which are not covered in this study, [. . .]"

(Line 174) "...the delay at a daily resolution for longer time periods than the one used in this study."

Answer: We will change the manuscript according to the suggestions.

(Line 175) The sentence is unclear and needs to be reworded. 38(Line 179) What do the authors mean by "global trend"? 39 (Line 179) The sentence is unclear and needs to be reworded.

Answer: We will clarify the sentences:

"The difference between the ionospheric delay for the European and Australian stations in Figures 7 and 8 show only small differences due to the assumed trend with the geomagnetic activity. This trend has to be removed in the further analysis. [. . .]The non-seasonal trends are removed by calculating the difference between the ionospheric delays of both stations."

(Line 183) "...a stronger seasonal variation..." 41 (Line188) "...with latitude in northern summer. The station at..." 42 (Line 193) remove "where data from high latitudes are
missing" because the Australian stations have a larger low-latitude extent than the European stations and this phrase does not reflect that. 43 (Line 194) recommend replacing "agree" with "are consistent" 44 (Line 197) "...good observational coverage..." 45 (Line 198) Remove repeated description of the IGS TEC map 46 (Line 202) "...Figure 11, which maps the mean delay values for the mid-latitudes in summer (May-August) and winter (November-February). Figure 11 shows delays that are consistent with the results from the European ionosonde stations (Figure 7b). 47 (Line 205) "...hours over the entire region."

Answer: We will change the manuscript according to the suggestions.

(Line 205) The sentence that begins at the end of this line is unclear and needs to be reworded.

Answer: We will clarify the sentence and add a reference to preceding studies to back up our discussion:

"The decrease of the ionospheric delay at latitudes greater than $65°$N and smaller $35°$N confirms a latitudinal trend, which was found in preceding studies (Lee et al., 2012)."

(Line 207) "...the delay decreases with increasing latitude. From..." 50 (Line 208) "...70âŮęN, or about -0.06 hours per degree in latitude." 51 (Line 211) "...is much smaller than the variation in latitude for the same season, with a change of..." 52 (Line 224) "...main analysis, we confirmed..." 53 (Line 234) Move the last two bullet points starting on this line to the previous paragraph where the authors were discussing the portions of previous studies that this study validated. 54 (Line 242) "Future analysis would benefit from high resolution ionospheric delay calculations for longer time periods that cover different..."

Answer: We will change the manuscript according to the suggestions.

(Line 243) Sentence starting at the end of this line is unclear and needs to be reworded

Answer: We will clarify the suggestion of including the thermospheric conditions in future analysis:

"The thermospheric conditions (e.g. neutral winds or composition changes in the atomic/molecular ratio), which are known for their impact on the ionosphere (Rishbeth, 1998; Rishbeth et al., 2000) should be included in future analysis as well."

(Data availability and acknowledgements) Not all acronyms are defined 57 (Line 297) "F 2" should be "F2" 58 (Line 327) page numbers missing and filled using n/a–n/a

Answer: We will change the manuscript according to the suggestions.

5. Referencing

Reference needed for the impact of solar irradiance on the vertical ionospheric structure

Answer: We will add the reference: M. Kelley, The Earth's Ionosphere, vol. 96. Academic Press, 2009.

(Line 44) Citation to a source that explains or demonstrated that TEC is dominated by the F2 peak response is needed

Answer: We will add the reference (see major comment 2): Elizabeth J. Petrie and Manuel Hernandez-Pajares and Paolo Spalla and Philip Moore and Matt A. King, "A Review of Higher Order Ionospheric Refraction Effects on Dual Frequency GPS," Surveys in Geophysics, vol. 32, no. 3, pp. 197–253, Nov. 2010

(Line 63) Which model is included in the TEC calculation? Include a very short description and a citation to this model.

Answer: We will add the references and move the model description to this line:

Orus, "Improvement of global ionospheric VTEC maps by using kriging interpolation technique," Journal of Atmospheric and Solar-Terrestrial Physics, vol. 67, no. 16, pp.

1598–1609, Nov. 2005.

M. Hernandez-Pajares et al., "Comparing performances of seven different global VTEC ionospheric models in the IGS context," in IGS Workshop 8-12 February 2016, 2016.

(Line 106) Citation for IGS TEC maps needed.

Answer: We will add the references for the IGS TEC maps.

(Line 196) Citation needed.

Answer: We will add the references:

C. Watson, P. T. Jayachandran, and J. W. MacDougall, "GPS TEC variations in the polar cap ionosphere: Solar wind and IMF dependence," Journal of Geophysical Research: Space Physics, vol. 121, no. 9, pp. 9030–9050, Sep. 2016.

N. Maruyama, "Dynamic and energetic coupling in the equatorial ionosphere and thermosphere," Journal of Geophysical Research, vol. 108, no. A11, 2003.

Please also note the supplement to this comment:
https://www.ann-geophys-discuss.net/angeo-2019-91/angeo-2019-91-AC2-supplement.zip
* * *
**Fig. 1.** Map of ionosonde stations with overlayed magnetic field configuration.

[Figure]

**Fig. 2.** Scatter plots for gradient of the ionospheric delay and Kp-index.

**Fig. 3.** Comparison of the ionospheric delay for Tromso and Rome.

---

## Referee Report (RR1)

The article **Spatial and seasonal effects on the delayed ionospheric response to solar EUV changes** presents a study that examines the relationship between solar EUV irradiance and F-region ionospheric density. This study builds off of previous work, confirming previous results using higher resolution calculations. It also examines seasonal and latitude variations for a small region of the globe. Both the validation efforts and the study into local European variations are of interest to the scientific community. The presentation and language are improved, but are not yet of high enough quality for publication. There are also significant errors in the physical reasoning that must be corrected, as they are based on a foundation that is demonstrably false. The length of the paper is adequate. Referencing is improved, but falls just short of appropriate. I believe this study could contribute positively to the scientific community if additional changes are made.

**1  Title and abstract**

The title is clear and appropriate. The abstract can be improved by addressing the following issues:

1. (Line 2) "...ionospheric response, testing and improving upon previous studies of this ionospheric delay. Several time series of correlation..."

2. (Line 3) " ..trend of the ionospheric delay from..."

3. (Line 7) "...region, the difference between..."

4. (Lines 7-8) Sentence is cumbersome and needs to be reworded.

5. (Lines 9-10) "...European region, and found to be characterised by a decrease in the delay from...at 70°N in the summer. For winter months, a roughly..."

6. (Line 11) "...summer months..."

7. (Lines 9-12) These two sentences repeat the same conclusion and should be consolidated. If the authors intended to impart something distinct in these two sentences, then they should be reworded.

8. (Line 13) "...also indicate that the ionospheric delay to EUV radiation depends on both geomagnetic activity and the 11-year solar cycle."

9. (Line 13) The abstract states that the results in this study support a variation with the 11-year solar cycle in the ionospheric delay, but there is not enough data to support this claim (much less than 11 years). The authors should adjust the wording in the abstract to match the more appropriate phrasing they used in the main text and conclusions.

**2 Major Issues and Questions**

1. The foF2 processing discussed at the end of Section 2 states that gaps are filled using a linear interpolation. What is the largest length of time allowed for the gaps?

2. The geomagnetic activity argument at the end of Section 3 states that the period of time considered in this study was during solar minimum. This is not true. This period of time begins during the ascending phase and ends during the main phase of the $24^{th}$ solar cycle (see the top panel of Figure 1 in this review, where the $F_{10.7}$ is plotted and the period of this study is noted by the dark red bars). The references and arguments of this section (e.g., Zieger and Mursula (1998)) need to be completely redone, since they start from a false assumption about the state of the solar activity level during this study.

3. Why is a weekly Kp index compared to an hourly ionospheric delay when it is more common to use a 3 hour Kp index? This unnecessary smoothing of the Kp index removes the motivation for using a high resolution ionospheric delay in this portion of the study and also reduces the perceived strength of geomagnetic activity (compare the bottom panel of Figure 1 in this review with Figure 4 (a) in the manuscript).

4. Figures need to be provided for the correlation between ionospheric delay and Kp for the southern hemisphere (something akin to Figure 5). This part of the analysis is needed to support the conclusions drawn on line 193.

5. Line 279 states that "better and more" EUV measurements are needed, but this is not presented as a deficit in this study (beyond the time range of available measurements). In what way do the EUV measurements need to be "better"? This should be stated in the main text, and the validity of this study's results placed in context of this observational deficit.

**3 Figures and tables**

1. (Figures 1, 11, 12): Red and green are a bad combination, as they are indistinguishable for many people suffering from colourblindness. Recommendations for different colours are available on sites such as:
https://ux.stackexchange.com/questions/94696/color-palette-for-all-types-of-color-blindness.
Figures can be tested for their appropriateness using sites such as:
https://www.color-blindness.com/coblis-color-blindness-simulator/.

[Figure]

Figure 1: F$_{10.7}$ and Kp Indices for the period of time used in this study.

**4 Grammar, referencing, and organisation**

1. (Lines 17-19) This sentence mentions several sources of EUV variability, but only provides a reference for solar flares. Referencing should encompass all of the mentioned sources of EUV variability either by citing multiple articles or by citing a reference article that covers all of the different topics.

2. (Lines 23-24) "A detailed understanding of the ionospheric...processes is needed to provide..."

3. (Lines 26, 27, 29, 100, 101, 103, 111, 113, 153, 255, 257, 259) Data is measured "at" different resolutions. For example, "data at a daily resolution", "data at an hourly resolution", or "data at hourly resolutions". Also note that "a" or "an" is only used before the singular conjugation of "resolution".

4. (Line 27-28) "...data at higher temporal resolution... of interest, as it permits more detailed descriptions of temporal and spatial variations."

5. (Line 28) "...can also be explored further."

6. (Line 29) "...delay at high temporal resolutions..."

7. (Line 37) "...and theoretical calculations were used..."

8. (Line 37-38) This sentence needs to be reworded to follow the standard grammatical structure "...the influence of $\mathbf{X}$ on $\mathbf{Y}$." As is, the sentence appears to be missing $\mathbf{Y}$ (though it is possible that I misunderstood the sentence and it is instead missing $\mathbf{X}$).

9. (Line 38) Remove comma between "both" and "the"

10. (Line 39) Recommend replacing "dominated" with "dominantly controlled"

11. (Lines 40-41) The transition between the last few sentences in this paragraph is jarring and should be rephrased.

12. (Line 46) "TEC measured the vertically integrated..."

13. (Lines 57-59) The sentences in this paragraph are constructed backwards. This paragraph should be reworded or cut entirely, as it isn't strictly necessary.

14. (Lines 61-62) "...with publicly available EUV observations provided by the Solar..."

15. (Line 63) "...and the Solar..."

16. (Line 66-67) "...represent almost the entire EUV spectrum, with a wavelength range from 0.1 to 105 nm, a spectral resolution of 0.1 nm, and a temporal resolution of 20 s. The EUV data cover several..."

17. (Line 74) "...which provide global coverage..."

18. (Lines 81, 85) "...derived from auto-scaled ionosonde..." (81) and cut "(auto-scaled)" on line (85)

19. (Line 89) "Instead, auto-scaled data from the..."

20. (Line 90) "...Canberra ionosondes are used for the analysis in the southern hemisphere."

21. (Line 91) "...are comparable, with a small magnetic..."

22. (Line 94) "...ionospheric parameter, foF2, measured with..."

23. (Line 95) "...mean foF2. Gaps..."

24. (Line 101) "...Table 1. The first delay..."

25. (Line 101) "...cross-correlations at an hourly resolution was performed by..."

26. (Line 102-104) "This work extends the previous research by addressing daily, seasonal, and regional dependencies of the ionospheric delay at a high temporal resolution . The analysis compares the ionospheric delay in the TEC and foF2 from different locations. Their corresponding time series are examined for different temporal variations, including: diurnal, 27-day solar rotation cycle, and seasonal."

27. (Line 106) "...daily, and hourly). The hourly resolution TEC data are extracted..."

28. (Line 106-107) "...(NASA, 2019b) at Rome..."

29. (Line 107) Are the times of day given in universal time, solar local time, or magnetic local time? Please specify in the text to avoid confusion.

30. (Line 112) "...same trend, though."

31. (Line 114) "...TEC is expected, as it is consistent with results from preceding studies (see Table 1). (Recommend starting a new paragraph here). Solar EUV radiation does not fully control the ionospheric variability..."

32. (Line 115) "...time scales, resulting in the low correlation coefficients shown in Figure 2 (b), (d), and (f) (Ungluab et al., 2012). The magnitude of the correlation coefficient has been shown to relate to the strength of the impact of other..."

33. (Line 116) "...Analyzing times of both high..."

34. (Line 117) "...ionospheric parameters is important to understand the changes in ionospheric processes and interactions."

35. (Line 120) "...90 days for the TEC..."

36. (Line 121) "...The two methods differ only in the way that the TEC time series was extracted from the..."

37. (Line 122) "...with a fixed location, the latitude..."

38. (Line 123) "...with a fixed local time, the longitude..."

39. (Line 123) "...the differences in the..."

40. (Line 130) "...the solar local time, and the calculated..."

41. (Lines 131-133) "...local times. This makes the fixed local time approach preferable for further analysis. However, its utility is limited since the time series extracted from the..."

42. (Line 133) "...on measurements (and more heavily on the background model) when considering areas..."

43. (Line 133-134) "...ground stations. Thus, this study preferentially utilises the fixed location method, since a location with good data coverage is more easily selected. And despite the strong diurnal..."

44. (Line 135) "...impact on both the correlation and the delay calculations."

45. (Line 135) "...at hourly..." (remove 'an')

46. (Line 137) "...they are of the same order..."

47. (Line 139) "...thermospheric conditions also impact the ionospheric state. During the period of this study (January 2011 through December 2013)..."

48. (Line 154) "...certain variations at longer time scales, while keeping..."

49. (Line 176-177) "...the ionospheric processes at this location..."

50. (Line 178-179) "...(Hansucker and Hargreaves, 2002). In this study, the station at Tromsø provides a high-latitude boundary for the analysis of the delayed ionospheric response in the European region."

51. (Line 180-181) "...general, the TEC and foF2 correlation coefficients at the Australian stations are slightly larger than the corresponding correlation coefficients at the European stations."

52. (Line 231) "...where good observational..."

53. (Line 232) "...stations and minimal influence..."

54. (Line 234) "...was done by calculating cross-correlations..."

55. (Line 234) "...of one hour, as shown in Figure..."

56. (Line 248) "The next analysis averages the calculated time series of delay maps over longitude..."

57. (Line 249) "...in Figure 14, and have a resolution..."

58. (Line 257) "...resolution through several different..."

59. (Line 258) "...fixed local times, fixed locations, and comparisons of correlation coefficients on different sub-annual time scales."

60. (Line 259) "...delay at high temporal resolutions."

61. (Line 279) "...conditions. Such work will require better and more abundant EUV measurements."

62. (Line 281) "...should also be included in future analysis. Results presented in this study need to be..."

63. (Line 283) "...this knowledge presents an opportunity..."

---

## Referee Report (RR2)

The article **Spatial and seasonal effects on the delayed ionospheric response to solar EUV changes** presents a study that examines the relationship between solar EUV irradiance and F-region ionospheric density. This study builds off of previous work, confirming previous results using higher resolution calculations. It also examines seasonal and latitude variations for a small region of the globe. Both the validation efforts and the study into local European variations are of interest to the scientific community. The presentation and language are greatly improved, with a few typos and awkward sentences present. The previous errors in the physical reasoning have also been corrected. The length of the paper is adequate. Referencing is appropriate. I believe this study contributes positively to the scientific community as is, though some minor changes would improve legibility.

**1  Title and abstract**

The title is clear and appropriate. The abstract is of an appropriate length and clearly summarises the scope and findings of the article.

**2  Figures and tables**

The figures and tables are clear, well labelled, and all necessary to support the article's text.

**3  Grammar and organisation**

1. (Line 45) "...analysis uses GNSS and ionosonde..."

2. (Line 52) "...TEC maps with good..."

3. (Line 60) "...EUV spectrum have been continuously..."

4. (Line 73) "...global coverage from 1998 onwards at the required..."

5. (Lines 86-87) "A complementary analysis of the southern hemisphere would preferentially use the South African region due to similarities in geographic longitude, but data gaps prevented a reliable estimation of the delay for the available stations."

6. (Line 89) "...used for the southern hemisphere analysis."

7. (Lines 91-92) "...These similarities make the selected stations appropriate for a comparison between the northern and southern hemispheres."

8. (Line 271) "...11-year solar cycle, or at least..."

9. (Line 289) "...Rishbeth et al., 2000), should also..."

---

## Author Response (AR2)

Thank you for all the comments and advice to our manuscript. We revised the manuscript according to your advice.

**Title and abstract**

1. (Line 2) "...ionospheric response, testing and improving upon previous studies of this ionospheric delay. Several time series of correlation..."

2. (Line 3) "...trend of the ionospheric delay from..."

3. (Line 7) "...region, the difference between..."

We changed the abstract according to the suggestions.

4. (Lines 7-8) Sentence is cumbersome and needs to be reworded.

We reworded the sentence: "The difference between northern and southern hemisphere is analyzed by comparisons with the Australian region. A seasonal variation of the delay between northern and southern hemisphere is calculated for TEC with ≈5±0.7 hours and foF2 with ≈8±0.8 hours."

5. (Lines 9-10) "...European region, and found to be characterised by a decrease in the delay from...at 70°N in the summer. For winter months, a roughly..."

6. (Line 11) "...summer months..."

We changed the abstract according to the suggestions.

7. (Lines 9-12) These two sentences repeat the same conclusion and should be consolidated. If the authors intended to impart something distinct in these two sentences, then they should be reworded.

We removed the second sentence.

8. (Line 13) "...also indicate that the ionospheric delay to EUV radiation depends on both geomagnetic activity and the 11-year solar cycle."

9. (Line 13) The abstract states that the results in this study support a variation with the 11-year solar cycle in the ionospheric delay, but there is not enough data to support this claim (much less than 11 years). The authors should adjust the wording in the abstract to match the more appropriate phrasing they used in the main text and conclusions.

We reworded the sentence: "Results also indicate a relation of the ionospheric delay to geomagnetic activity and a possible correlation with the 11-year solar cycle in the analyzed time period."

**Major Issues and Questions**

1. The foF2 processing discussed at the end of Section 2 states that gaps are filled using a linear interpolation. What is the largest length of time allowed for the gaps?

The mean length of a data gap is ≈2 hours for the chosen data sets.  The standard deviations vary for each station, e.g. Rome with ≈7 hours and Canberra with ≈14 hours. The longest consecutive data gap for Rome is 13 hours. For Canberra longer data gaps (several days) appear, but there is no delay calculated at these time periods (see the data gaps in the ionospheric delay, e.g. in Figure 9).

We added a clarification: "Delay calculations during data gaps of several days do not succeed due to the lack of a defined peak in the cross-correlation. This causes corresponding gaps in the observed trend of the ionospheric delay." (Line 94-96)

2. The geomagnetic activity argument at the end of Section 3 states that the period of time considered in this study was during solar minimum. This is not true. This period of time begins during the ascending phase and ends during the main phase of the 24th solar cycle (see the top panel of Figure 1 in this review, where the F10:7 is plotted and the period of this study is noted by the dark red bars). The references and arguments of this section (e.g., Zieger and Mursula (1998)) need to be completely redone, since they start from a false assumption about the state of the solar activity level during this study.

We changed the section and address the special conditions during the ascending phase of the 24th solar cycle: "Geomagnetic activity and thermospheric conditions also impact the ionospheric state. The period of this study (January 2011 through December 2013) covers the ascending phase and beginning of the main phase of the 24th solar cycle. The geomagnetic activity during this time is on very low levels compared to previous ascending phases with geomagnetic storm rates that compare to solar minima in previous cycles (Richardson, 2013). The solar activity of the cycle is also significantly lower compared to previous cycles and a much weaker ionization of the ionosphere occurs (Hao et al., 2014). These complex variations are not covered by EUV flux measurements and cannot be characterized with the cross-correlations between solar EUV and ionospheric parameters." (Line 141-147)

3. Why is a weekly Kp index compared to an hourly ionospheric delay when it is more common to use a 3 hour Kp index? This unnecessary smoothing of the Kp index removes the motivation for using a high resolution ionospheric delay in this portion of the study and also reduces the perceived strength of geomagnetic activity (compare the bottom panel of Figure 1 in this review with Figure 4 (a) in the manuscript).

We changed Figure 4 to show the Kp-index without smoothing.

For the scatter plots with the Kp-index and the ionospheric delay in Figure 5 and 6 a weekly resolution is used to remove the strong variations at shorter time scales. The plots compare the long term changes (semiannual and annual variation) of Kp-index and ionospheric delay and, therefore, the chosen smoothing is appropriate. An analysis of more immediate changes of the ionospheric delay with geomagnetic activity changes is not aim of the analysis and should be done with different approaches in future studies.

4. Figures need to be provided for the correlation between ionospheric delay and Kp for the southern hemisphere (something akin to Figure 5). This part of the analysis is needed to support the conclusions drawn on line 193.

We added Figure 6 showing the corresponding scatter plots for Canberra. In 2011 and 2012 good correlations are observed. There is no correlation in 2013, which is due to the strong deviations of the calculated delay during the end of the year. Nevertheless, both hemispheres show a clear correlation between geomagnetic activity and ionospheric delay.

5. Line 279 states that "better and more" EUV measurements are needed, but this is not presented as a deficit in this study (beyond the time range of available measurements). In what way do the EUV measurements need to be "better"? This should be stated in the main text, and the validity of this study's results placed in context of this observational deficit.

We changed the part: "Such work will require ongoing efforts to measure the solar EUV radiation in the future, since these data are the basis for the delay calculations." (Line 286-287)

**Figures and tables**

(Figures 1, 11, 12): Red and green are a bad combination, as they are indistinguishable for many people suffering from colourblindness. Recommendations for different colours are available on sites such as: https://ux.stackexchange.com/questions/94696/color-palette-for-all-typesof-color-blindness. Figures can be tested for their appropriateness using sites such as: https://www.color-blindness.com/coblis-color-blindness-simulator/.

We adjusted the plots. Lines are changed to a combination of red and blue.

**Grammar, referencing, and organisation**

1. (Lines 17-19) This sentence mentions several sources of EUV variability, but only provides a reference for solar flares. Referencing should encompass all of the mentioned sources of EUV variability either by citing multiple articles or by citing a reference article that covers all of the different topics.

We add references for the topics (including the next sentence):

Solar cycle - C. Fröhlich and J. Lean, "Solar radiative output and its variability: evidence and mechanisms," The Astronomy and Astrophysics Review, vol. 12, no. 4, pp. 273–320, Dec. 2004.

27-day solar rotation cycle - J. L. Lean et al., "Solar extreme ultraviolet irradiance: Present, past, and future," Journal of Geophysical Research: Space Physics, vol. 116, no. A1, p. n, Jan. 2011.

Processes in ionosphere - H. Rishbeth and M. Mendillo, "Patterns of F2-layer variability," Journal of Atmospheric and Solar-Terrestrial Physics, vol. 63, no. 15, pp. 1661–1680, Oct. 2001.

2. (Lines 23-24) "A detailed understanding of the ionospheric...processes is needed to provide..."

3. (Lines 26, 27, 29, 100, 101, 103, 111, 113, 153, 255, 257, 259) Data is measured "at" different resolutions. For example, "data at a daily resolution", "data at an hourly resolution", or "data at hourly resolutions".

Also note that "a" or "an" is only used before the singular conjugation of "resolution".

4. (Line 27-28) "...data at higher temporal resolution... of interest, as it permits more detailed descriptions of temporal and spatial variations."

5. (Line 28) "...can also be explored further."

6. (Line 29) "...delay at high temporal resolutions..."

7. (Line 37) "...and theoretical calculations were used..."

We changed the manuscript according to the suggestions.

8. (Line 37-38) This sentence needs to be reworded to follow the standard grammatical structure "...the influence of X on Y." As is, the sentence appears to be missing Y (though it is possible that I misunderstood the sentence and it is instead missing X).

We changed the part: "Simulations with the Thermosphere Ionosphere Electrodynamics General Circulation Model (TIEGCM) and theoretical calculations were used to discuss the influence of ion production and loss on the ionospheric delay. The impact of the $O/N_2$ ratio on the delay was analyzed as well."

9. (Line 38) Remove comma between "both" and "the"

10. (Line 39) Recommend replacing "dominated" with "dominantly controlled"

We changed the manuscript according to the suggestions.

11. (Lines 40-41) The transition between the last few sentences in this paragraph is jarring and should be rephrased.

We changed the transition: "The resulting ionospheric response could be further modulated by dynamic and electrodynamic processes in the ionosphere. In addition, a latitudinal dependence of the ionospheric delay was shown (Ren et al., 2018)." (Line 41-43)

12. (Line 46) "TEC measured the vertically integrated..."

13. (Lines 57-59) The sentences in this paragraph are constructed backwards. This paragraph should be reworded or cut entirely, as it isn't strictly necessary.

14. (Lines 61-62) "...with publicly available EUV observations provided by the Solar..."

15. (Line 63) "...and the Solar..."

16. (Line 66-67) "...represent almost the entire EUV spectrum, with a wavelength range from 0.1 to 105 nm, a spectral resolution of 0.1 nm, and a temporal resolution of 20 s. The EUV data cover several..."

17. (Line 74) "...which provide global coverage..."

18. (Lines 81, 85) "...derived from auto-scaled ionosonde..." (81) and cut "(auto-scaled)" on line (85)

19. (Line 89) "Instead, auto-scaled data from the..."

20. (Line 90) "...Canberra ionosondes are used for the analysis in the southern hemisphere."

21. (Line 91) "...are comparable, with a small magnetic..."

22. (Line 94) "...ionospheric parameter, foF2, measured with..."

23. (Line 95) "...mean foF2. Gaps..."

24. (Line 101) "...Table 1. The first delay..."

25. (Line 101) "...cross-correlations at an hourly resolution was performed by..."

26. (Line 102-104) "This work extends the previous research by addressing daily, seasonal, and regional dependencies of the ionospheric delay at a high temporal resolution. The analysis compares the ionospheric delay in the TEC and foF2 from different locations. Their corresponding time series are examined for different temporal variations, including: diurnal, 27-day solar rotation cycle, and seasonal."

27. (Line 106) "...daily, and hourly). The hourly resolution TEC data are extracted..."

28. (Line 106-107) "...(NASA, 2019b) at Rome..."

We changed the manuscript according to the suggestions.

29. (Line 107) Are the times of day given in universal time, solar local time, or magnetic local time? Please specify in the text to avoid confusion.

The times refer to local time. We added a clarification to the text: "The daily and weekly data sets for TEC are retrieved by calculating the corresponding means for the values from 11:00 to 13:00 local time each day, i.e. only the time periods with an expected maximum photoionization are considered." (Line 108-110)

30. (Line 112) "...same trend, though."

31. (Line 114) "...TEC is expected, as it is consistent with results from preceding studies (see Table 1). (Recommend starting a new paragraph here). Solar EUV radiation does not fully control the ionospheric variability..."

32. (Line 115) "...time scales, resulting in the low correlation coefficients shown in Figure 2 (b), (d), and (f) (Ungluab et al., 2012). The magnitude of the correlation coefficient has been shown to relate to the strength of the impact of other..."

33. (Line 116) "...Analyzing times of both high..."

34. (Line 117) "...ionospheric parameters is important to understand the changes in ionospheric processes and interactions."

35. (Line 120) "...90 days for the TEC..."

36. (Line 121) "...The two methods differ only in the way that the TEC time series was extracted from the..."

37. (Line 122) "...with a fixed location, the latitude..."

38. (Line 123) "...with a fixed local time, the longitude..."

39. (Line 123) "...the differences in the..."

40. (Line 130) "...the solar local time, and the calculated..."

41. (Lines 131-133) "...local times. This makes the fixed local time approach preferable for further analysis. However, its utility is limited since the time series extracted from the..."

42. (Line 133) "...on measurements (and more heavily on the background model) when considering areas..."

43. (Line 133-134) "...ground stations. Thus, this study preferentially utilizes the fixed location method, since a location with good data coverage is more easily selected. And despite the strong diurnal..."

44. (Line 135) "...impact on both the correlation and the delay calculations."

45. (Line 135) "...at hourly..." (remove 'an')

46. (Line 137) "...they are of the same order..."

47. (Line 139) "...thermospheric conditions also impact the ionospheric state. During the period of this study (January 2011 through December 2013)..."

48. (Line 154) "...certain variations at longer time scales, while keeping..."

49. (Line 176-177) "...the ionospheric processes at this location..."

50. (Line 178-179) "...(Hansucker and Hargreaves, 2002). In this study, the station at Tromsø provides a high-latitude boundary for the analysis of the delayed ionospheric response in the European region."

51. (Line 180-181) "...general, the TEC and foF2 correlation coefficients at the Australian stations are slightly larger than the corresponding correlation coefficients at the European stations."

52. (Line 231) "...where good observational..."

53. (Line 232) "...stations and minimal influence..."

54. (Line 234) "...was done by calculating cross-correlations..."

55. (Line 234) "...of one hour, as shown in Figure..."

56. (Line 248) "The next analysis averages the calculated time series of delay maps over longitude..."

57. (Line 249) "...in Figure 14, and have a resolution..."

58. (Line 257) "...resolution through several different..."

59. (Line 258) "...fixed local times, fixed locations, and comparisons of correlation coefficients on different sub-annual time scales."

60. (Line 259) "...delay at high temporal resolutions."

61. (Line 279) "...conditions. Such work will require better and more abundant EUV measurements."

62. (Line 281) "...should also be included in future analysis. Results presented in this study need to be..."

63. (Line 283) "...this knowledge presents an opportunity..."

We changed the manuscript according to the suggestions. Thank you again for all the helpful advice and detailed corrections!

[revised manuscript text omitted]